# Wildfire precursors show complementary predictability in different timescales

Yuquan Qu [1] ✉, Diego G. Miralles [2], Sander Veraverbeke [3], Harry Vereecken [1] & Carsten Montzka [1]

In most of the world, conditions conducive to wildfires are becoming more prevalent. Net carbon emissions from wildfires contribute to a positive climate feedback that needs to be monitored, quantified, and predicted. Here we use a causal inference approach to evaluate the influence of top-down weather and bottom-up fuel precursors on wildfires. The top-down dominance on wildfires is more widespread than bottom-up dominance, accounting for 73.3% and 26.7% of regions, respectively. The top-down precursors dominate in the tropical rainforests, mid-latitudes, and eastern Siberian boreal forests. The bottom-up precursors dominate in North American and European boreal forests, and African and Australian savannahs. Our study identifies areas where wildfires are governed by fuel conditions and hence where fuel management practices may be more effective. Moreover, our study also highlights that top-down and bottom-up precursors show complementary wildfire predictability across timescales. Seasonal or interannual predictions are feasible in regions where bottom-up precursors dominate.

Fire weather is becoming more severe because of climate change[1-3] and is expected to increase the number, extent, and severity of wildfires in many regions on Earth[4,5]. For example, the 2019/20 Black Summer wildfires in Australia were initiated and intensified by consecutive unprecedented dry years[6], the unprecedented 2021 Siberia wildfires followed record-breaking heatwave and drought[7], and the 2021 British Columbia wildfires followed widespread Northwest Pacific and Canada heatwave[8]. Wildfires play an important role in ecosystem dynamics and species preservation[9,10]. They can be beneficial to some ecosystems since smaller wildfires can prevent catastrophic wildfires by consuming fuels, or even promote species replacement and enhance their adaptation to climate change[11,12]. However, despite these positive effects, wildfires often have a negative impact on the environment and society, for example, through increased forest loss, carbon emissions, and smoke exposure[13,14]. Moreover, they contribute to climate warming by reinforcing a positive feedback loop[15,16]. In boreal ecosystems, this feedback loop may shift the carbon balance from sink to source[17].

Better understanding and management of wildfires, therefore, are necessary to mitigate their negative impacts on human livelihood and the Earth's natural system. The availability of fuel is highly influenced by previous and current vegetation status and is a precondition for wildfires, yet extensive burning only occurs when fuels are dry and weather conditions are conducive to wildfire spread[18]. The Canadian Forest Wildfire Weather Index (FWI) system[19] is widely used in wildfire studies to indicate fuel moisture, wildfire spread, fuel availability, and wildfire intensity by integrating temperature, relative humidity, wind speed, and precipitation over different time lags. The FWI system was originally developed for boreal ecosystems, although it has been used more widely[3]. However, the wildfire drivers vary at a global scale, and this variability is not fully understood because of the lack of mechanistic knowledge of ecosystem-specific wildfire precursors (or predictors).

Here, we apply a causal discovery algorithm considering lagged influences to detect the relationships between top-down weather and bottom-up fuel precursors and wildfires (in terms of burned area per climate and land cover class). This causal discovery algorithm, the Peter & Clark Momentary Conditional Independence (PCMCI)[20,21], uses delayed time series to automatically and conditionally select relevant

[1]Institute of Bio- and Geosciences: Agrosphere (IBG-3), Forschungszentrum Jülich GmbH, Jülich, Germany. [2]Hydro-Climate Extremes Lab, Ghent University, Ghent, Belgium. [3]Faculty of Science, Vrije Universiteit Amsterdam, Amsterdam, Netherlands. ✉e-mail: y.qu@fz-juelich.de

variables from high dimensional data sets and enable a meaningful causal interpretation[21]. We focus on investigating relationships between global wildfire occurrence, more specifically quantified in terms of burned area (BA), and wildfire precursors, for the period 2003–2020. The wildfire precursors were separated into two groups: (i) top-down group, including maximum air temperature (Tmax), vapor pressure deficit (VPD), potential evaporation (ET0), wind speed (Wind), and aridity anomaly index (AAI); (ii) bottom-up group, including the fraction of photosynthetically active radiation (FPAR), gross primary production (GPP), normalized difference vegetation index (NDVI), enhanced vegetation index (EVI), and soil water deficit index (SWDI). The details and selection of these precursors can be found in the Methods and Supplementary Methods.

## Results

### Spatial pattern of dominant precursors for wildfires

To investigate the dominant wildfire precursors globally, we applied PCMCI to build causal networks for each of the 232 ecoregions (29 climate zones multiplied by 8 vegetation types), using the top-down weather and bottom-up fuel grouping system. Then, in each ecoregion, we may detect a dominant precursor belonging to either the top-down or bottom-up group according to whether there is a causal relationship and the strength of the causal relationship (in this case partial correlation). High wildfire predictability is found in southeast North America, the tropical rainforests, the Sahel of Africa, the Congo

Basin, the East European Plain and West Siberian Plain, the eastern Central Siberian Plateau, southeast China, and central and western Australia (Fig. 1a). Globally, the top-down group shows more dominance than the bottom-up group, accounting for 73.3% and 26.7% of the globe where causal relationships are detected (Fig. 1c). The bottom-up dominance is found mainly in the North American and European boreal forests, western North America, the African and Australian savannahs, and eastern South America (Fig. 1b). The other regions are dominated by the top-down group.

We ranked the importance (according to the dominant area fraction/pixel number) of the precursors in each group (Supplementary Fig. 1). In the top-down group, ET0 is the most important precursor followed by VPD, AAI, Tmax, and Wind, accounting for 41.1%, 32.0%, 16.6%, 10.3%, and 0% of top-down dominant regions, respectively. In the bottom-up group, SWDI is the most important precursor followed by GPP, EVI, FPAR, and NDVI, accounting for 56.4%, 21.4%, 11.1%, 8.6%, and 2.5% of bottom-up dominant regions, respectively. Wind speed is usually believed to play an important role in wildfire spread, while we detected almost no dominance globally. There could be two reasons: (1) wind speed gains dominance only when wildfires are ignited, while our study involved both fire season and non-fire season. A similar example is wildfires in the western United States are decoupled from lighting activities unless weather and vegetation conditions are considered[22]; (2) wind can both promote and suppress wildfire[23], and this opposite effect diminishes its dominance.

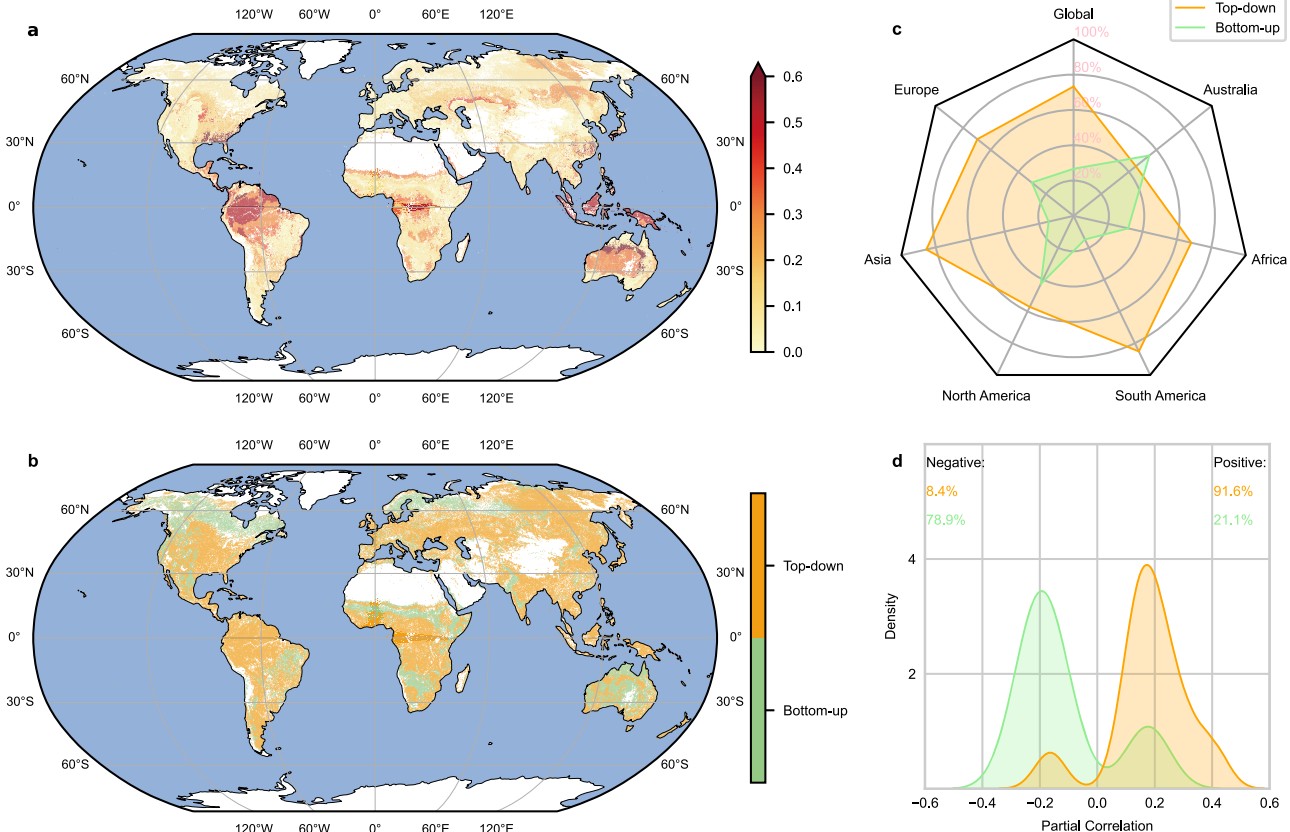

**Fig. 1 | The dominant precursor group of global wildfires.** The causal analysis was performed on 29 climate zones and 8 vegetation types (232 ecoregions in total) with a 0.05 significance level. **a** The sum of squares of the partial correlation coefficients of all 10 precursors to wildfires per ecoregion, with higher values meaning higher predictability of burned area, and vice versa. **b** The group (top-down or bottom-up) with the highest partial correlation in each ecoregion. White areas in **a** and **b** are where causal relationships are not detected or insignificant. **c** The dominant area fraction of each precursor group globally and on continental scales. **d** The probability density distribution of partial correlations for each group. For each group, only the partial correlations where this group is dominant (higher absolute value of partial correlation than another group) are shown. The positive or negative fractions of the partial correlations for each group are also shown (top right and top left) to help explain the causality. Note that the area, not the height (y-axis value), represents the probability. Source data are provided as a Source Data file.

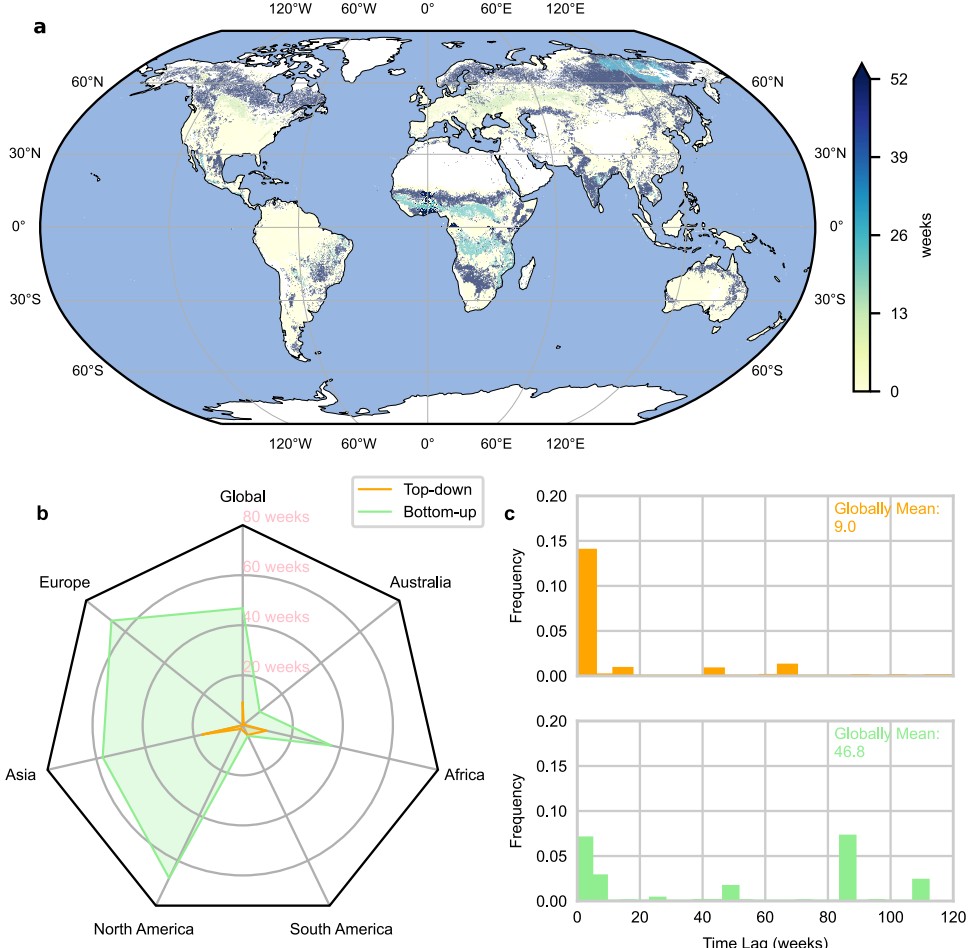

**Fig. 2 | Time lags between wildfires and their precursors. a** The spatial pattern of weighted mean time lags. The time lags were weighted per ecoregion by partial correlations of all the 10 precursors to wildfires. **b** The weighted mean time lags between wildfire and each group globally and on continental scales. The time lag for each group was calculated from the area-weighted mean where this group is dominant. **c** The dominant time lag distribution of the precursor groups. For each group, only the time lags when this precursor group is dominant are shown. Source data are provided as a Source Data file.

The distribution of partial correlations for the two groups is shown in Fig. 1d. Wildfires are positively related to top-down precursors globally (91.6% positive). This positive relationship indicates that in flammability-limited regions, BA will increase when more energy is supplied (higher Tmax and ET0) or the environment is drier (higher VPD and AAI). This is supported by evidence indicating that wildfires increase with the increased frequency and intensity of summer heatwaves[24], the strengthened solar radiation absorption[25,26], and the increased atmospheric water demand and drought conditions[27,28] (see Supplementary Fig. 2 for ET0, AAI, and VPD dominance). Major climate modes (e.g., El Niño–Southern Oscillation, and North Atlantic Oscillation) also modulate the occurrence of these weather anomalies[29–33] and as such play an important role in governing global patterns of wildfire activities[32,34,35]. The positive top-down dominance in the rainforests is supported by studies showing that here wildfires are limited to the time window when fuels are dry enough to burn[36,37], see Supplementary Fig. 2 for the AAI dominance in the tropical rainforests.

Wildfires tend to be negatively related to bottom-up precursors (78.9% negative, Fig. 1d). Negative patterns prevail in boreal forests, the Brazilian Highlands, southeastern China, Australia, and parts of African savannahs (Supplementary Fig. 5) supporting that vegetation growth limiting factors (such as droughts, heatwaves, and vermin outbreaks) from the previous year or pre-fire season may suppress vegetation growth and maintain fire-prone conditions of fuels[38,39]. Positive patterns are found mostly in drier and temperate regions (e.g., the western United States, African savannahs, northwestern India, and western Europe, Supplementary Fig. 5), confirming wildfires there are limited by fuels, not flammability. In these regions, wildfires will increase when more fuels are provided (higher FPAR, GPP, NDVI, and EVI) or there is more water supply to increase fuel accumulation by a higher plant growth rate (higher SWDI).

## Time lags between precursors and wildfires

For wildfire predictions, the lag between the time when wildfire precursors prevail and a wildfire event is important information because appropriate wildfire-favoring environments often need time to develop. Here, the mean time lags weighted by partial correlations between wildfires and precursors are investigated in detail. Longer time lags are shown in the boreal forests, African and Australian savannahs, the Indian Peninsula, the Indochina Peninsula, and southeastern South America (Fig. 2a). In the other regions, wildfires are a direct result of the precursing conditions (time lags mostly no more than 1 week).

On a global and continental scale, the time lags of top-down precursors are always shorter than bottom-up precursors (Fig. 2b). These short time lags can be explained by the higher variability of top-down conditions and their direct influence on wildfires (not through changing the fuel accumulation). Longer time lags of top-down precursors are limited in Eastern Siberia, the Indian Peninsula, the

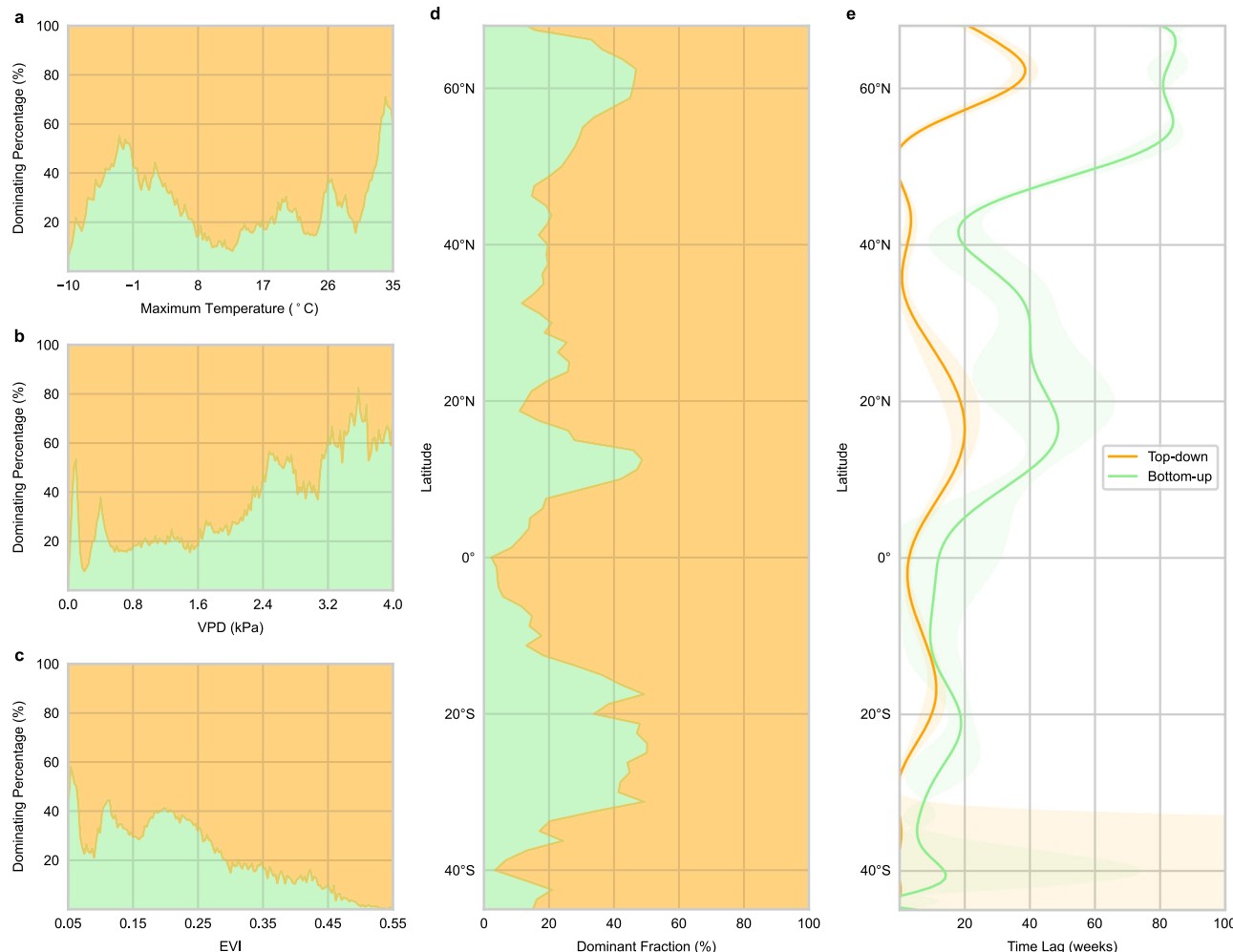

**Fig. 3 | The dominance and time lags of wildfire precursors with maximum temperature, vapor pressure deficit (VPD), enhanced vegetation index (EVI), and latitude.** The dominant fraction that is primarily driven by the group (top-down or bottom-up), is binned by maximum temperature, VPD (higher values mean drier conditions), EVI, and latitude (**a**–**d**). The fractions were calculated by dividing the number of pixels where a specific precursor group is dominant in a specific bin by the number of pixels located in this bin and dominated by any of the two groups. The sum of the fractions of the two groups in each bin is 100%. **e** For each group, the weighted mean time lag in each bin was area-weighted by all the time lags located in this bin and when this group is dominant. The x-axis for maximum temperature, VPD, EVI, and the y-axis for latitude were all divided into 200 bins. Source data are provided as a Source Data file.

Indochina Peninsula, eastern South America, and tropical Africa (Supplementary Fig. 6). In this case, one possible reason is that the top-down precursors affect wildfires through their influence on fuel accumulation and availability.

The bottom-up precursors show long time lags, especially in Europe, Asia, North America, and Africa (more than half a year, Fig. 2b). One possible reason could be that the current vegetation status or accumulated fuel is related to water supply and vegetation growth in the pre-wildfire seasons[39,40]. This is supported by studies showing that spring greening related to global warming can either be positively or negatively related to total biomass[41,42], through the carryover effect of current vegetation states on subsequent growth caused by increased photosynthesis or earlier soil moisture depletion[38,39]. The bottom-up precursors can also come with negligible time lags, mainly in Australia and South America (Fig. 2b and Supplementary Fig. 7), indicating that they can serve as not only fuel accumulation (longer time lag) but also fuel dryness (short time lag) indicators. Note that Fig. 2c shows the dominant time lag distribution in each group, we also investigated the weighted mean time lag distribution (no matter whether the group is dominant or not). Interestingly, the weighted mean case (Supplementary Fig. 11) shows that the time lags of bottom-up precursors are usually short, with a peak of around no more than one week and a

second peak of around 60 weeks. We assume that bottom-up precursors serve as fuel accumulation indicators when they dominate wildfires, and fuel dryness indicators when top-down precursors dominate wildfires.

## What controls the dominance of wildfire precursors?
To generalize the previous analysis, we evaluated the dominance changes (according to the dominant pixel number) of top-down and bottom-up precursors that are primarily driven by energy, dryness, vegetation, and location (Tmax, VPD, EVI, and latitude, Fig. 3). The dominance of bottom-up precursors peaks when Tmax is around −3°C and 32°C, in regions where fuels are limited or take a longer time to accumulate. When Tmax is between −3°C and 32°C, the top-down precursors gain dominance as fuels may be more readily available. The dominance of top-down precursors decreases when dryness (higher VPD) and sparseness of vegetation (lower EVI) rise. In arid regions, wildfires are limited by fuel accumulation and connectivity. This corroborates the positive relationship between bottom-up precursors and wildfires in dry regions like western North America and African savannahs (Supplementary Fig. 5). When the environment is generally wet (lower VPD) and fuels are not limited (higher EVI), top-down precursors gain dominance, as can be seen in the positive relation

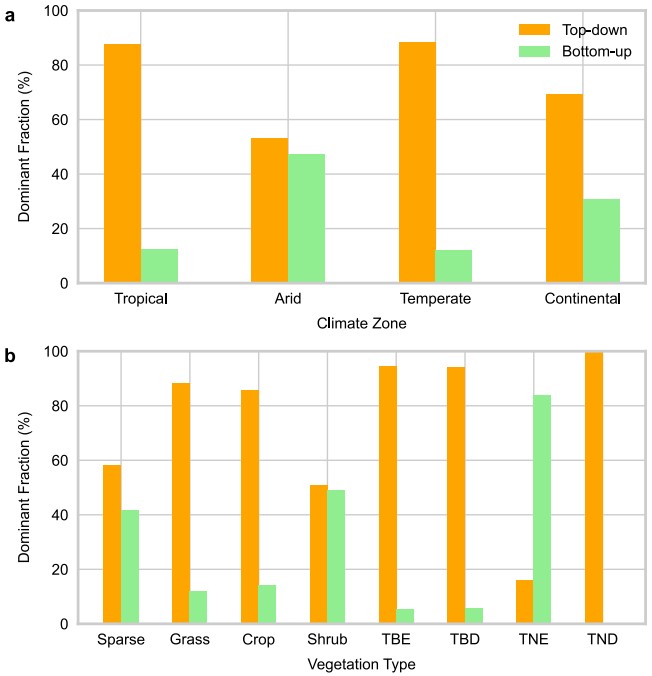

**Fig. 4 | The dominance of the top-down and bottom-up groups according to climate zones and vegetation types.** The dominant fractions were calculated by dividing the dominant pixels of each group by the total number of pixels dominated by any of the two precursor groups in climate zones (**a**), and vegetation types (**b**). There are five main climate zones on Earth: tropical, arid, temperate, continental, and polar. Here we only focused on the first four climate zones because there is only one ecoregion in the Polar zone with limited wildfires. The eight vegetation types include sparse vegetation, grassland, cropland, shrubland, tree cover broad-leaf evergreen (TBE), tree cover broad-leaf deciduous (TBD), tree cover needle-leaf evergreen (TNE), tree cover needle-leaf deciduous (TND). Source data are provided as a Source Data file.

between top-down precursors and fire activity (Supplementary Fig. 4). This is supported by the study conducted by Qing et al., 2022[43] which indicates that flash droughts which promote wildfires caused by heatwave or precipitation deficiency (top-down conditions) are more likely to occur in humid and semi-humid regions.

On a global scale, the top-down precursors are more dominant in the tropical rainforests, mid-latitudes, and boreal forests (eastern Siberia) where the weather is humid, and fuels are abundant or less flammable (Fig. 1b and Fig. 3d). In tropical rainforests, the environmental conditions are not suitable for wildfires unless top-down conditions are extreme[44]. Therefore, top-down precursors almost synchronize with fire activity in these regions[38,45] (Fig. 3e). In the mid-latitudes of Eurasia, the top-down dominance may be caused by sufficient fuels and reliable dry seasons. Moreover, as shown in Supplementary Fig. 15, the fuel type in the mid-latitudes mainly consists of fine fuels, which will dry out in a short time under hot and dry weather conditions[46]. In North America, the top-down precursors dominate wildfires through, e.g., earlier winter snowpack melting caused by temperature increase, droughts related to summer transpiration increase[40,47], and plant growth or photosynthesis changes influenced by VPD through stomatal closure[48]. The dominance of top-down precursors in eastern Siberia boreal forests is supported by the study conducted by Kim et al., 2020[49] which shows that in southeastern Siberian boreal forests, the length of the wildfire season is temperature-limited. Moreover, here the changes in top-down conditions may alter thaw depth[50] and cause early snow-melting above the frozen soil, but this melted water converts to surface runoff and cannot supply soil and vegetation. This process and the low-pass filtering effect of soils[51] (i.e., filtering out high-frequency variations of

precipitation and showing longer persistence) may explain the long time lag of top-down precursors (Fig. 3e).

Three peaks of bottom-up dominance show around 60°N, 10°N, and 20°S which corresponds to North American and European boreal forests, and African and Australian savannahs (Fig. 1b and Fig. 3d). Note that wildfires in North American and European boreal forests show bottom-up dominance while in eastern Siberian boreal forests show top-down dominance. This could be explained by the fact that in eastern Siberia, boreal forests consist of fire-resistant species, and wildfires there are prone to low-intensity surface fires, while in North American boreal forests where fire-embracing black spruce forests are dominant, wildfires are prone to high-intensity crown fires which consume more fuels[52]. A study conducted by Alvarado et al., 2020[53] indicated that wildfires in Australian savannahs are mainly controlled by fuel load, while in African savannahs they are equally controlled by fuel load and fuel dryness. This may explain the overwhelming bottom-up dominance in Australian savannahs and approximately equal dominance of top-down and bottom-up precursors in African savannahs (Fig. 1b). In African savannahs, evidence also shows that precipitation reduction and cropland expansion reduce fuel accumulation, connectivity, and flammability, and thus prevent the ignition and spread of wildfires[54–56], as can be seen in the positive partial correlations of bottom-up precursors in this region (Supplementary Fig. 5).

We further investigated the influence of climate zone and vegetation type on the dominance of wildfire precursors, the results are shown in Fig. 4. The top-down precursors show more dominance in tropical, temperate, and continental zones than in arid zones. This is supported by the top-down dominance in Fig. 3a, b when the climate is cold and humid. In arid zones, including sparse vegetation and shrubland, the dominance of top-down precursors is weaker as wildfires are limited by fuels, which is in line with Fig. 3b (high VPD). In grassland, the top-down precursors show dominance as the seasonal grass dies every year and turns to dead fine fuels, whose dryness quickly responds to changes in top-down precursors[46]. In cropland where the fuel load is more stable, wildfires are more likely controlled by top-down precursors. The top-down precursors show more dominance in broad-leaf forests than in needle-leaf forests. Needle leaves generally have lower water content and are more flammable than broad leaves, which makes the ignition and spread of wildfires less limited by top-down precursors. In needle-leaf forests, especially needle-leaf evergreen forests, wildfires are coupled with bottom-up precursors as needle leaves decompose slower than broad leaves[57]. Wildfires in broad-leaf evergreen forests are more affected by top-down precursors. For example, in the Amazon rainforests, the forests have generally large and bulky leaves, which makes them less flammable, and wildfires tend to be surface fires while dead fuels cannot accumulate because of the high decomposition rate[58], decoupling the wildfires from bottom-up precursors.

## Discussion

Globally, top-down weather dominance on wildfire activity is more widespread than bottom-up fuel dominance. As bottom-up conditions can be directly influenced by wildfire management decisions[59], our precursor attribution highlights areas where fuel management is able to effectively mitigate wildfires. Fuel management could be effective where the bottom-up conditions dominate wildfires, while it will likely be less influential where wildfires are dominated by the top-down conditions[60]. Our results show that fuel management may be more efficient in mitigating wildfires in North American and European boreal forests, and African and Australian savannahs. Removing or altering fuels in these regions may increase landscape heterogeneity and reduce wildfire spread[61]. Outside of these bottom-up dominant areas, building large fuel breaks, mechanical thinning, and controlled burning are also increasingly being applied e.g., in temperate forests of

North America, the Mediterranean, and Australia[3]. However, in tropical rainforests, mid-latitudes, and eastern Siberian boreal forests where the weather is humid, fuels are abundant, or less flammable, wildfires are largely decoupled from the bottom-up status making fuel management less efficient[60,62].

Fuel monitoring and management become more important in future climate change scenarios. Drier and warmer conditions may suppress fuel accumulation and likely convert the fire regimes from flammability-limited to fuel-limited[37]. In Amazonia, deforestation changes the local weather system at forest edges which facilitates the replacement of rainforests with flammable herbaceous species[37]. In the eastern Siberian boreal forests, global warming gives rise to the invasion of more flammable tree species and thereby increases the probability of wildfires[63]. In boreal forests, climate warming increases lightning activities and the occurrence of overwintering fires, and hence the number of large wildfires would likely increase if these fire starts are combined with sufficient fuels[37,64]. Prescribed burning before the fire season potentially reduces wildfire risk in bottom-up dominant African and Australian savannahs[65]. However, this type of fuel management strategy is challenging under global warming, as the safe weather window suitable for prescribed burning is shortened[66]. Similarly, attention should also be paid to fire management strategies like long-term fire suppression, as the consequent excessive fuel accumulation increases the risk of high-intensity wildfires, especially in North American boreal forests where wildfires are dominated by bottom-up conditions[15].

There are three limitations of this study that could be addressed further. First, although benefiting from the big data era, the number of precursors included in this study is still limited. Major climate modes (e.g., El Niño–Southern Oscillation, and North Atlantic Oscillation) modulate regional top-down and bottom-up conditions and dominant wildfires[67,68]. Therefore, wildfire prediction could benefit from the achievable predictability of El Niño–Southern Oscillation[69] and other major climate modes. Human behavior (e.g., ignition, cropland expansion, and fire suppression) is also an important wildfire impact factor that may alter causal relationships. However, limited by the availability of high temporal resolution climate mode and human behavior indices, we currently cannot conduct such an analysis. Second, a better wildfire precursor grouping system may be beneficial. While we divided the precursors into top-down and bottom-up groups which are conceptually clear[70], sometimes, the top-down precursors dominate wildfires through their influence on fuel accumulation that is deduced from their long time lags, making the causal inference complex. Third, we assumed the causal relationship to be constant throughout the whole study period. However, extreme events or human behavior can alter fire regimes and potentially change causal relationships.

In conclusion, this is the first globally consistent study partitioning top-down weather and bottom-up fuel precursors of wildfire activity. On a global scale, top-down weather precursors dominate in 73.3% of regions where causal relationships are detected. In the remaining 26.7% of regions, bottom-up fuel precursors are dominant, which coincides with North American and European boreal forests, and African and Australian savannahs. In these regions, fuel management practices may be the most efficient. In addition to exploring wildfire causal relationships, we also focused on time lags between wildfires and their precursors. The time lag identified in this study is the time delay when wildfire precursors and BA show the highest partial correlation (highest predictive power). This information is helpful for wildfire prediction by implying that (1) we cannot make reliable wildfire predictions beyond these time lags in advance; (2) when tracing the causes of a wildfire that has already occurred, we should at least consider the time delay as time lags show, otherwise, we will miss critical information. Our study also highlights that global wildfire precursors show complementary timescales of predictability: bottom-up precursors show long-term predictability serving as fuel

accumulation indicators; top-down precursors show contemporary or short-term predictability serving as direct fire weather indicators. Seasonal or interannual wildfire prediction is feasible in regions like the boreal forests, African and Australian savannahs, the Indian Peninsula, the Indochina Peninsula, and southeastern South America when bottom-up precursors are considered.

## Methods

### Datasets

The wildfire indicator used in this study is the burned area (BA) taken from FireCCI51[71]. FireCCI51 is based on spectral and thermal information from the Moderate Resolution Imaging Spectroradiometer (MODIS) near-infrared band and active wildfire products. It is a global and monthly BA product with two different spatial resolutions (250 m of pixel level and 0.25° of grid level). Here we used the pixel-level data. The main information is included in three layers, i.e., the first fire detection day, the confidence level, and its corresponding land cover. A quality control process was applied to mask low-quality BA pixels whose confidence level is lower than 70%. A logarithmic transformation was applied to transform skewed BA data to a normal Gaussian distribution to make the detected causal relationship reliable[72,73].

The top-down precursors include maximum air temperature (Tmax), vapor pressure deficit (VPD), potential evaporation (ET0), wind speed (Wind), and aridity anomaly index (AAI). The bottom-up precursors include the fraction of photosynthetically active radiation (FPAR), gross primary production (GPP), normalized difference vegetation index (NDVI), enhanced vegetation index (EVI), and soil water deficit index (SWDI). A detailed description of the precursors - including which products were used, their spatial and temporal resolution, how they were calculated, and how these precursors were selected - can be found in the Supplementary Methods.

All the above-mentioned data sets span from 2003 to 2020. We performed the causal analysis at the ecoregional level considering that top-down weather and bottom-up fuel conditions are relatively similar within the same ecoregion[74]. We believe that this spatial aggregation is appropriate to generalize findings at a global scale, yet still allows for sufficient spatial variation when analyzing global fire regimes. In addition, we can generate a "look-up table" based on ecoregions that provide basic insights needed to develop a wildfire warning/predicting system for locations where we do not have enough in-situ or field data. In each of the 232 ecoregions (29 climate zones and 8 land cover classes), BA and precursors were resampled to a spatial resolution of 0.25° and averaged (precursors) or summed up (BA) to weekly values. Then, we calculated the ecoregional mean (precursors) or sum (BA) values by averaging or summing up all the pixels located in the ecoregions. Note that even though FireCCI51 is a monthly dataset, it includes a layer recording the first fire detection day. Therefore, we computed the weekly summed BA from this daily information. For the precursors, we calculated the weekly mean from their original temporal scale.

The climate zone data used in this study is the global digital Köppen-Geiger map[75] from the University of East Anglia and the German Weather Service representing the climate classification of the period from 1986 to 2010 at the spatial resolution of 5 arc minutes. The vegetation type used in this study is from the land cover layer of FireCCI51.

### Causal Analysis

The Peter & Clark Momentary Conditional Independence (PCMCI)[20,21] is implemented in this study. It is based on time series analysis which takes lagged variables into account to detect spatiotemporal dependency. It consists of two stages: a Peter & Clark (PC) condition selection stage and a Momentary Conditional Independence (MCI) test stage. In the first stage, for each variable, the irrelevant conditions that do not pass the iterative independence test are removed based on a Markov set discovery algorithm. In the next stage, a false-positive control was adapted

to highly relevant interdependent conditions from the first stage. In applications, two main assumptions of PCMCI are causal sufficiency and time series stationarity. The first assumption can be satisfied by considering as many relevant variables as possible. Normally, causal discovery is more reliable when more variables are included. But when too many irrelevant variables are included, the dimensionality increases, and the effect sizes become smaller. The above-mentioned condition selection and condition independence test stages in PCMCI mitigate this causal discovery dilemma[21]. The second assumption can be approximately satisfied by removing the seasonal cycle and trend in the time series. However, real-world applications can violate this assumption when the seasonal cycle is not fixed like some climate modes such as El Niño-Southern Oscillation. Experiments show that PCMCI is efficient even when this assumption is violated, e.g., the time series is highly autocorrelated or has a nonstationary trend[21]. We aimed to achieve (a) causal sufficiency by involving a wild range of wildfire precursors, and (b) time series stationarity by detrending and calculating anomalies. We used linear regression to remove the trend over the entire period and calculated anomalies to remove the seasonal cycle and alleviate temporal autocorrelation. The formula for the anomaly calculation can be found in the Supplementary Methods. We used a linear condition independence test, partial correlation, in PCMCI (both the condition selection and condition independence test stages) and set a significance level of 0.05 and a maximum time lag of 156 weeks. In each of the 232 ecoregions, we input wildfire activity indicator (BA), top-down precursors (Tmax, VPD, ET0, Wind, and AAI), and bottom-up precursors (FPAR, GPP, NDVI, EVI, and SWDI) and applied PCMCI to get causal networks showing causal links, directions, strengths, and time lags. In this study, we only focused on causal links and the time lags from wildfire precursors to BA. In each ecoregion, after getting the causal network, only the precursors that show causal links to BA are ranked by causal strength (partial correlation), and the precursor ranked first is called the dominant precursor, and its time lag is called the dominant time lag.

## Data availability
All relevant data generated in this study have been deposited in the Zenodo database under accession code https://doi.org/10.5281/zenodo.8421505. Source data are provided with this paper.

## Code availability
All codes used in this work for causal inference and data analysis are available from the corresponding authors upon request.

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

## Acknowledgements

This study is supported by the China Scholarship Council (CSC) under grant number 201906040220 (Y.Q.), the Deutsche Forschungsgemeinschaft (DFG, German Research Foundation) – SFB

1502/1-2022 – project number 450058266 (Y.Q., C.M., and H.V.), the Dutch Research Council through Vidi grant 016.Vidi.189.070 and the European Research Council through a Consolidator grant under the European Union's Horizon 2020 research and innovation program under grant agreement No. 101000987 (S.V.), and the European Research Council under grant agreement 101088405 (HEAT, D.G.M.).

## Author contributions

Y.Q. conceived the study and wrote the paper under the supervision of C.M. and H.V.; Y.Q. and C.M. designed and performed the study; D.G.M. and S.V. contributed interpretations of the causal results; Y.Q., D.G.M., S.V., H.V., and C.M. edited and approved the paper.

## Funding

## Competing interests

The authors declare no competing interests.
