## [Peer Review File · Nature Communications]

Wildfire Precursors Show Complementary Predictability in Different TimescalesREVIEWER COMMENTS

Reviewer #1 (Remarks to the Author):

General comments

This study focused on identifying the most important local predictors of fires. The authors provided comprehensive analysis on multiple aspects of the impacts of fire precursors. I find the results interesting and helpful. However, several issues should be addressed. I have listed my comments below.

Major comments

- This work focused on the impacts of local predictors on fires. However, the method is partly unclear. For example, for a given pixel with fire data, is the data of predictors also obtained from the same pixel or surroundings? As fires may not be limited in one pixel, the selected approach may affect the results and conclusions.
- ENSO is a single useful predictor of fires in many regions (e.g., North America, the tropics, Australia (Fang et al., 2021; Fuller & Murphy, 2006)). In fact, ENSO predictability is achievable (Ham et al., 2019) and the variations of most of fire precursors are driven by ENSO (Le & Bae, 2022; Thirumalai et al., 2017). Hence, it might be too complicated to use multiple fire precursors as suggested by this work.
- Similar to the above concern, I think major climate modes (e.g., ENSO and NAO) can be dominant fire precursors, in case these modes are included in the analyses.

Minor comments

- Lines 84-96: The role of ENSO should be mentioned. I find this study (Le et al., 2022) might be relevant as it showed that ENSO can be a single predictor of fire over multiple regions. In fact, ENSO modulate regional atmospheric precursors (e.g., surface temperature (Thirumalai et al., 2017), evaporation (Le & Bae, 2020)), hydrologic precursors (e.g., soil moisture (Le & Bae, 2022)) and vegetation precursors (e.g., GPP (Le et al., 2021)).
- Lines 225-251: The role of major climate modes should be discussed. For example, ENSO (McPhaden et al., 2006) and NAO (Hurrell et al., 2003) modulate the variations of multiple fire predictors and thus may affect regional fires.

References

- Fang, K., Yao, Q., Guo, Z., Zheng, B., Du, J., Qi, F., et al. (2021). ENSO modulates wildfire activity in China. *Nature Communications*, 12(1), 1764. <https://doi.org/10.1038/s41467-021-21988-6>
- Fuller, D. O., & Murphy, K. (2006). The Enso-Fire Dynamic in Insular Southeast Asia. *Climatic Change*, 74(4), 435–455. <https://doi.org/10.1007/s10584-006-0432-5>
- Ham, Y. G., Kim, J. H., & Luo, J. J. (2019). Deep learning for multi-year ENSO forecasts. *Nature*, 573(7775), 568–572. <https://doi.org/10.1038/s41586-019-1559-7>
- Hurrell, J. W., Kushnir, Y., Ottersen, G., & Visbeck, M. (2003). An overview of the North Atlantic Oscillation. In *Geophysical Monograph American Geophysical Union* (pp. 1–35). American Geophysical

Union. <https://doi.org/10.1029/134GM01>

Le, T., & Bae, D.-H. (2020). Response of global evaporation to major climate modes in historical and future Coupled Model Intercomparison Project Phase 5 simulations. *Hydrology and Earth System Sciences*, 24(3), 1131–1143. <https://doi.org/10.5194/hess-24-1131-2020>

Le, T., & Bae, D. (2022). Causal Impacts of El Niño–Southern Oscillation on Global Soil Moisture Over the Period 2015–2100. *Earth’s Future*, 10(3). <https://doi.org/10.1029/2021EF002522>

Le, T., Ha, K., & Bae, D. (2021). Increasing Causal Effects of El Niño–Southern Oscillation on the Future Carbon Cycle of Terrestrial Ecosystems. *Geophysical Research Letters*, 48(24). <https://doi.org/10.1029/2021GL095804>

Le, T., Kim, S., & Bae, D. (2022). Decreasing causal impacts of El Niño–Southern Oscillation on future fire activities. *Science of The Total Environment*, 826, 154031. <https://doi.org/10.1016/j.scitotenv.2022.154031>

McPhaden, M. J., Zebiak, S. E., & Glantz, M. H. (2006). ENSO as an integrating concept in earth science. *Science (New York, N.Y.)*, 314(December), 1740–1745. <https://doi.org/10.1126/science.1132588>

Thirumalai, K., DInezio, P. N., Okumura, Y., & Deser, C. (2017). Extreme temperatures in Southeast Asia caused by El Niño and worsened by global warming. *Nature Communications*, 8, 1–8. <https://doi.org/10.1038/ncomms15531>

Reviewer #2 (Remarks to the Author):

REVIEWER REPORT

SUMMARY

In this study, the authors used a causal inference approach (Peter & Clark Momentary Conditional Independence, PCMCi) to evaluate the influence of atmospheric, hydrologic, and vegetation precursors on fires and their time lags.

GENERAL COMMENTS

My overall appreciation is that the study is very interesting and has the advantage of being performed at a global scale, but the manuscript has too many flaws and needs a major revision before can be accepted for publication. The most important problems are lack of clarity and detail, especially in what concerns data and methods; vague, obvious, redundant, and unnecessary sentences; conceptual and methodological issues. Another problem with this manuscript is that it is not clear what is its contribution to increasing knowledge about wildfire factors. The authors have to explain which of their results are new/innovative, i.e., what are the noteworthy results?

Next, I will address what seems to me to be the main problems with the study/manuscript and then list a set of specific questions/comments/suggestions that help explain my overall appreciation.

Confidence in the results and conclusions strongly depends on the quantity and quality of the data as well as the adequacy of the methodology. The description of data and methodologies needs to be clarified and detailed. The reader does not always understand what the authors did, which prevents them from perceiving and validating results and conclusions. In this regard, one important issue is the air

temperature (Temp) precursor. The authors do not explain which temperature (minimum, mean, maximum, other) and at what level (surface, 2 m, 10 m, 850 hPa, etc.). Based on the statement that "All the data sets are averaged or summed up to weekly..." maybe we can assume that Temp is some kind of average air temperature. However, it is also common knowledge that the vast majority of wildfires and burned areas occur under conditions more extreme than average. The authors somehow acknowledge this fact in the first paragraph of the "Main" section. Thus, in addition to the authors having to explain which temperature they used, if they used the mean temperature, they will have to explain why. The results of such a study strongly depend on the chosen precursors. A bad choice of precursors of a given type/group leads to equally bad results for that type of precursor. How can the reader be confident that the selected precursors are the best/most suitable? It is also not clear how the authors selected the precursors. Was there a pre-selection process? If yes, which one? How did they conclude that four of each type would be the proper number?

If I understand correctly, this study has serious conceptual (please see specific comments) and methodological flaws that can decisively affect the results and lead the reader to wrong conclusions. One of the most important conceptual/methodological problems is the names and composition of the precursors' groups. First, there are misclassified precursors. For example, the group of hydrological precursors does not seem to me to be the most suitable, as half of them are not hydrological, but atmospheric (precipitation and FWI). See examples referred to in specific comments. Secondly, and as a consequence of the first, the groups do not seem to me to be the most suitable. For example, in the case of hydrological precursors, it seems to me that it would be more appropriate to consider a group of soil precursors or, possibly, soil moisture precursors. On the one hand, the state and type of vegetation depend more on soil moisture than on all other hydrological aspects. I conclude that the authors agree with this statement because the group of hydrological precursors integrate measurements of soil moisture and not hydrology (e.g., surface and groundwater, with regard to their mechanical, physical and chemical properties, their geographic distribution, etc.). On the other hand, precursors should be independent within each group and between groups, which they clearly are not. The variables are all related to each other in such an obvious way that it is not even necessary to mention an example. The wildfire research community recognizes and agrees that, for a vegetation wildfire to occur, the following requirements are necessary: (i) fuels/plant biomass, (ii) atmospheric conditions that lead the fuels/vegetation to an adequate state of dryness and, (iii) a source of ignition. The adequate state of dryness of the fuels (live or dead vegetation) is reached with atmospheric conditions of high temperature and low humidity that lead the vegetation to hydric and/or thermal stress for some time (depending on the type and state of the vegetation).

Despite this common knowledge, the authors decided to reclassify atmospheric factors into atmospheric and hydrological. They consider: measures of energy (temperature, radiation and evaporation) and call them atmospheric precursors; and, measures of water availability (precipitation, air and soil humidity), which are essentially atmospheric and call them hydrological precursors. It is not possible to conclude that the hydrologic precursors are more or less important than the others if the hydrologic precursors are not hydrologic.

The authors also do not describe what might be the limitations and constraints that could affect the conclusions of the study.

Without a major revision of the manuscript to clarify and correct the problems identified in this general appreciation and in the specific comments, it is not worthwhile to analyse other aspects of the manuscript.

SPECIFIC COMMENTS

Line 4: In this manuscript, the authors used the concept of fire (a planned, wanted and controlled combustion), which I believe is not correct; the authors should use the concept of wildfire (an unplanned and unwanted combustion of dead or alive vegetal biomass in a rural or natural area).

Line 4: the first sentence is completely vague; what role? What services? How?

Line 30: Is it really possible to "predict" wildfires "reliably and accurately

Lines 31-32: "Fuel availability, usually in the form of vegetation conditions,...", fuel availability is not equal to vegetation conditions. Please correct the sentence.

Line 36: I agree that the FWI system is used widely. However, since it was developed for Canada, it must be calibrated before can be used in any other location. How did you calibrate the FWI indices in order to be able to use them globally?

Line 46: I suppose that the black circle character is a typo/not necessary.

Line 46: you mentioned that you use air temperature (Temp). However, you do not explain what air temperature is; the minimum, mean, maximum or other air temperature? Air temperature at what level?

Line 48: can you explain why did you classify Precipitation, which can be defined as the fall of water, in any form (rain, snow, hail, hail, etc.), from the atmosphere to the earth's surface, as a hydrologic and not an atmospheric precursor?

Lines 48-49: Please explain why you classify the fire WEATHER index (FWI) as a hydrologic and not an atmospheric precursor. Please note that the Canadian Forest Fire Weather Index (FWI) System consists of six components that account for the effects of fuel moisture and WEATHER conditions on fire behaviour, computed with noon air temperature, air relative humidity and wind speed, and 24-hour accumulated precipitation.

Please quantify and explain how these misclassifications impact your findings.

Line 57 (Fig. 1a): Please explain why the sum of the partial correlations of all 12 fire precursors per ecoregion ranges between 0 and 2.

Lines 57-61:

"For all the 232 ecoregions, the dominant area fraction of the precursor (Fig. S1) shows that VPD (47.6%) is the most important atmospheric precursor followed by Rad, Temp, and ETO (23.7%, 18.1%, 10.6%, respectively)". What is the meaning of these results? Does this mean that VPD is the most important precursor in 47.6% of the 232 ecoregions, analysed independently or jointly?

If I understand correctly, the sum of the percentages of the four precursors of each type is 100%. Does this mean that you apply PCMCI independently for the precursors of each type/group (for all the ecoregions)? Please clarify.

Lines 72-73: How did you compute these percentages? Did you use the number of ecoregions, the area, or what?

Line 83 and 91: this is only due to the precursors you selected and how you classify them.

Line 87: another black circle!

Lines 87-89: An obvious and unnecessary sentence. Is there any region where the atmospheric precursors do not play an important role?

Line 91-92: "Fires tend to be negatively related to hydrologic precursors (72.6%), and less frequently this relationship is positive (27.4%), see Fig. 1d"; the part ", and less frequently this relationship is positive (27.4%)," of the sentence is completely unnecessary; the reader can easily compute 100-

72.6=27.4.

Lines 92-94: another obvious and unnecessary sentence; it is obvious that drought and precipitation deficits may provide optimal conditions for wildfires while the opposite, a “high water supply” can prevent fires.

Lines 94-96: What “dominant positive relationship”

Line 110: I am confused. If “All the data sets are averaged or summed up to weekly” (line 287), how can you have assessed the partial correlation for time lags shorter than 1 week?

Lines 112-113: “..., which means in such regions fires are ignited shortly after (within a few days) the changes in the fire precursors.” another obvious and unnecessary sentence.

Lines 118-119: what evidence do you have for such a statement? That should not be the reason, at least everywhere.

Lines 125-127: this does not seem correct/accurate; the tropics are a very large region; it is possible to be much more accurate. The explanation does not also seem correct. Please note that 30 weeks is less than a year.

Line 140 and 154: area of what?

Line 154: another black circle.

Line 165: another black circle.

Line 196: in the Köppen-Geiger climate classification, the B type of climate is arid, not dry. Please correct.

Lines 254-270 (data):

Apparently, you used monthly fire data can the reader assume that the time series of all other variables are also monthly? If not (line 287), how did you proceed to analyse data with different temporal sampling?

Datasets do not have the same spatial resolution; how did you process the data with different spatial resolutions?

Lines 262-263. Why did you use temperature, radiation, and precipitation of the Global Land Data Assimilation System (GLDAS) but ET₀, VPD, FWI and AAI from ERA5? Why did you not use all weather data from the same source, for example, ERA5, which provides you with not only global but also consistent (homogeneous) data?

Lines 271-285: the reader may not have access to all publications cited in the manuscript; this is why the methodology must be, eventually briefly but clearly described, which is not the case. In addition, the description of how AAI was computed is unclear; for example, “For each year, we calculated the difference between AI at a certain period and the normal AI for the corresponding period to calculate the anomaly, then the AAI was obtained.”

Lines 286-305 (Data Processing Procedures):

you mentioned that you use monthly fire data (line 256 “...It is a global and monthly BA product”) and “All the data sets are averaged or summed up to weekly values...”. Please explain this apparent contradiction.

So, you produce weekly time series for each of the 28 climate zones (from Köppen-Geiger) and 8 land cover classes (from ESA CCI). The questions are:

What was the original temporal resolution of the downloaded data?

How did you proceed to obtain the time series for each of the 232 ecoregions 28 (or 29) climate zones and 8 vegetation types?

Line 294: Please clarify what is a “... PC condition selection ...”; all acronyms must be clearly defined.

Lines 294-295: Please clarify what is “...an MCI conditional independence test ...”; all acronyms must be

clearly defined.

Line 295: "In the first stage, irrelevant conditions are removed"; what is an irrelevant condition?

Lines 296-298: "Two main assumptions of PCMCI are time-series stationarity and causal sufficiency. The first assumption can be satisfied by calculating anomalies and de-trending the time series." Once again, the authors do not adequately explain what leads to doubt. First, obviously, it is not by calculating anomalies that the series becomes stationary; just think that the trend does not disappear. Second, you don't explain what trend you remove and how. Did you just remove the trend over the entire study period? Did you check and correct the seasonality? This whole procedure must be properly explained.

Reviewer #1 (Remarks to the Author):

General comments

This study focused on identifying the most important local predictors of fires. The authors provided comprehensive analysis on multiple aspects of the impacts of fire precursors. I find the results interesting and helpful. However, several issues should be addressed. I have listed my comments below.

Response: We appreciate the constructive comments and suggestions provided by the reviewer to improve the quality of our manuscript. Our responses to the reviewer's major and specific comments are given below.

Major comments

Major comment #1

- This work focused on the impacts of local predictors on fires. However, the method is partly unclear. For example, for a given pixel with fire data, is the data of predictors also obtained from the same pixel or surroundings? As fires may not be limited in one pixel, the selected approach may affect the results and conclusions.

Response: We appreciate the reviewer's careful review and revised the Methods part to clarify how we processed the data. Since not the entire Earth surface has experienced wildfires in the past decades, we performed the analysis based on 232 ecoregions. We, therefore, implemented the following approach: We first resampled all the data to a spatial resolution of 0.25° and calculated the weekly mean (wildfire precursors) or sum (burned area) values. Then, we calculated the ecoregion mean or sum values by averaging or summing up all the pixels located in the ecoregions, i.e., the following analysis is purely based on the ecoregion level, not the pixel level. The revised Methods are as follows:

The wildfire indicator used in this study is the BA taken from FireCCI151⁷³. FireCCI151 is based on spectral and thermal information from the Moderate Resolution Imaging Spectroradiometer (MODIS) near-infrared (NIR) band and active wildfire products. It is a global and monthly BA product with two different spatial resolutions (250m of pixel level and 0.25° of grid level). Here we used the pixel-level data. The main information is included in three layers, i.e., the first fire detection day, the confidence level, and its corresponding land cover. A quality control process was applied to mask low-quality BA pixels whose confidence level is lower than 70%. A logarithmic transformation was applied to transform skewed BA data to a normal Gaussian distribution to make the detected causal relationship reliable^{74, 75}.

The top-down precursors include maximum air temperature (Tmax), vapor pressure deficit (VPD), potential evaporation (ET0), wind speed (Wind), and aridity anomaly index (AAI). The

bottom-up precursors include the fraction of photosynthetically active radiation (FPAR), gross primary production (GPP), normalized difference vegetation index (NDVI), enhanced vegetation index (EVI), and soil water deficit index (SWDI). A detailed description of the precursors - including which products were used, their spatial and temporal resolution, how they were calculated, and how these precursors were selected - can be found in the Supplementary Materials.

All the above-mentioned data sets span from 2003 to 2020. Since not the entire world surface has experienced wildfires in the past decades, we performed the analysis based on ecoregions instead of at a pixel level. In each of the 232 ecoregions (29 climate zones and 8 land cover classes), BA and precursors were resampled to a spatial resolution of 0.25° and averaged (precursors) or summed up (BA) to weekly values. Then, we calculated the ecoregional mean (precursors) or sum (BA) values by averaging or summing up all the pixels located in the ecoregions. Note that even though FireCCI51 is a monthly dataset, it includes a layer recording the first fire detection day. Therefore, we computed the weekly summed BA from this daily information. For the precursors, we calculated the weekly mean from their original temporal scale.

The climate zone data used in this study is the global digital Köppen-Geiger map⁷⁶ from the University of East Anglia and the German Weather Service representing the climate classification of the period from 1986 to 2010 at the spatial resolution of 5 arc minutes. The vegetation type used in this study is from the land cover layer of FireCCI51. **(Line 286-313)**

Major comment #2

- ENSO is a single useful predictor of fires in many regions (e.g., North America, the tropics, Australia (Fang et al., 2021; Fuller & Murphy, 2006)). In fact, ENSO predictability is achievable (Ham et al., 2019) and the variations of most of fire precursors are driven by ENSO (Le & Bae, 2022; Thirumalai et al., 2017). Hence, it might be too complicated to use multiple fire precursors as suggested by this work.

Response: We appreciate the reviewer's insights about the dominance of major climate modes, like ENSO, on wildfires, and we agree. Part of our author team is involved in an ongoing study, in which the influence of major climate modes on burned area is investigated. Below are all the climate mode indices we included in the ongoing study:

Climate Mode	Climate Mode Index
Antarctic Oscillation	AAO index (daily)
Arctic Oscillation	AO index (daily)
Polar Eurasia Pattern	PEA index (monthly)
North Atlantic Oscillation	NAO index (daily)
East Atlantic West Russia Pattern	EAWR index (monthly)
Tropical Southern Atlantic Dipole	TSA index (monthly)
Tropical Northern Atlantic Dipole	TNA index (monthly)
Indian Ocean Dipole	IOD index (monthly)
West Pacific Pattern	WP index (monthly)
Pacific–North American Pattern	PNA index (daily)
El Niño Southern Oscillation	ENSO index (daily)

The majority of climate mode indices are available at a monthly temporal resolution. In our current study, we are interested in the influences of precursors on a weekly temporal resolution, and hence it is not feasible to integrate the climate indices in this framework. A weekly temporal resolution is desirable for our study because fire activity can fluctuate on a daily scale, and much of this information would be lost at a monthly resolution.

Although we did not involve climate modes in this study, our results and findings are influential and complementary to earlier work that investigated relationships between climate modes and fire activity. Climate modes act upon wildfires through their influence on the top-down and bottom-up predictors that are here investigated (e.g., precipitation, temperature, GPP).

In conclusion, our current precursor study has its own focus. This causal study focuses on what dominates wildfires, where, and the time lags between wildfire precursors and wildfires. For the abovementioned reasons, we believe that the inclusion of the climate modes would distract from the main storyline of this manuscript and would go beyond its scope.

We provide further discussion on relationships between our study and earlier studies focusing on relationships between climate modes and wildfires as suggested by the reviewer in Minor comment #1 and Minor comment #2 (see below).

Major comment #3

- Similar to the above concern, I think major climate modes (e.g., ENSO and NAO) can be dominant fire precursors, in case these modes are included in the analyses.

Response: Thanks for the comment. As we responded in Major comment #2, the inclusion of climate modes will not change the dominance of top-down or bottom-up precursors, and we consider this complementary information. We have appended a discussion on how major climate modes may influence precursors and fire activity, see Minor comment #1 and Minor comment #2.

Minor comments

Minor comment #1

- Lines 84-96: The role of ENSO should be mentioned. I find this study (Le et al., 2022) might be relevant as it showed that ENSO can be a single predictor of fire over multiple regions. In fact, ENSO modulate regional atmospheric precursors (e.g., surface temperature (Thirumalai et al., 2017), evaporation (Le & Bae, 2020)), hydrologic precursors (e.g., soil moisture (Le & Bae, 2022)) and vegetation precursors (e.g., GPP (Le et al., 2021)).

Response: Thanks for the reviewer's constructive suggestion. We addressed the role of major climate modes on wildfires as follows:

Wildfires are positively related to top-down precursors globally (91.6% positive, Fig. 1d). This positive relationship indicates that in flammability-limited regions, BA will increase when more

energy is supplied (higher T_{max} and ET_0) or the environment is drier (higher VPD and AAI). This is supported by evidence indicating that wildfires increase with the increased frequency and intensity of summer heatwaves²⁵, the strengthened solar radiation absorption^{26,27}, and the increased atmospheric water demand and drought conditions^{28, 29} (see Fig. S2 for ET_0 , AAI , and VPD dominance). Major climate modes (e.g., El Niño–Southern Oscillation, and North Atlantic Oscillation) also modulate the occurrence of these weather anomalies^{30, 31, 32, 33, 34} and as such play an important role in governing global patterns of wildfire activities^{33, 35, 36}. The top-down dominance in the rainforest is supported by studies showing that here wildfires are limited to the time window when fuels are dry enough to burn^{37, 38}, see Fig. S2 for the AAI dominance. **(Line 86-97)**

Reference:

30. Le T, Bae DH. Causal Impacts of El Niño–Southern Oscillation on Global Soil Moisture Over the Period 2015–2100. *Earth's Future* 2022, 10(3).

31. Le T, Bae D-H. Response of global evaporation to major climate modes in historical and future Coupled Model Intercomparison Project Phase 5 simulations. *Hydrology and Earth System Sciences* 2020, 24(3): 1131-1143.

32. Le T, Ha KJ, Bae DH. Increasing Causal Effects of El Niño–Southern Oscillation on the Future Carbon Cycle of Terrestrial Ecosystems. *Geophysical Research Letters* 2021, 48(24).

33. Le T, Kim SH, Bae DH. Decreasing causal impacts of El Niño–Southern Oscillation on future fire activities. *Sci Total Environ* 2022, 826: 154031.

34. Thirumalai K, DiNezio PN, Okumura Y, Deser C. Extreme temperatures in Southeast Asia caused by El Niño and worsened by global warming. *Nat Commun* 2017, 8: 15531.

35. Fang K, Yao Q, Guo Z, Zheng B, Du J, Qi F, et al. ENSO modulates wildfire activity in China. *Nat Commun* 2021, 12(1): 1764.

36. Fuller DO, Murphy K. The ENSO–Fire Dynamic in Insular Southeast Asia. *Climatic Change* 2006, 74(4): 435-455.

Minor comment #2

- Lines 225-251: The role of major climate modes should be discussed. For example, ENSO (McPhaden et al., 2006) and NAO (Hurrell et al., 2003) modulate the variations of multiple fire predictors and thus may affect regional fires.

Response: We appreciate the reviewer's constructive comment. As explained in our response to Major comment #2, we find that our study on precursors influences global fire activity complementary to earlier work that investigated relationships between major climate modes and fire activity. In our revision, we now better connect our work with these earlier studies and discuss as a limitation the non-explicit consideration of the influence of climate modes:

There are three limitations of this study that could be addressed further. First, although benefiting from the big data era, the number of precursors included in this study is still limited. Major climate modes (e.g., El Niño–Southern Oscillation, and North Atlantic Oscillation) modulate regional top-down and bottom-up conditions and dominant wildfires^{69, 70}. Therefore, wildfire prediction could benefit from the achievable predictability of El Niño–Southern Oscillation⁷¹ and other major climate modes. Human behavior (e.g., ignition, cropland expansion, and fire suppression) is an important wildfire impact factor that may alter causal relationships. However, limited by the availability of high temporal resolution climate mode and human behavior indices, we currently cannot conduct such an analysis. Second, a better wildfire precursor grouping system may be beneficial. While we divided the precursors into top-down and bottom-up groups which are conceptually clear⁷², sometimes, the top-down precursors dominate wildfires through their influence on fuel accumulation that is deduced from their long time lags, making the causal inference complex. Third, we assumed the causal relationship to be constant throughout the whole study period. However, extreme events or human behavior can alter fire regimes and potentially change causal relationships. (Line 253-266)

Reference:

69. Hurrell JW, Kushnir Y, Ottersen G, Visbeck M. An Overview of the North Atlantic Oscillation. *The North Atlantic Oscillation: Climatic Significance and Environmental Impact*, 2003, pp 1-35.

70. McPhaden MJ, Zebiak SE, Glantz MH. ENSO as an integrating concept in earth science. *Science* 2006, 314(5806): 1740-1745.

71. Ham YG, Kim JH, Luo JJ. Deep learning for multi-year ENSO forecasts. *Nature* 2019, 573(7775): 568-572.

References

Fang, K., Yao, Q., Guo, Z., Zheng, B., Du, J., Qi, F., et al. (2021). ENSO modulates wildfire activity in China. *Nature Communications*, 12(1), 1764. <https://doi.org/10.1038/s41467-021-21988-6>

Fuller, D. O., & Murphy, K. (2006). The Enso-Fire Dynamic in Insular Southeast Asia. *Climatic Change*, 74(4), 435–455. <https://doi.org/10.1007/s10584-006-0432-5>

Ham, Y. G., Kim, J. H., & Luo, J. J. (2019). Deep learning for multi-year ENSO forecasts. *Nature*, 573(7775), 568–572. <https://doi.org/10.1038/s41586-019-1559-7>

Hurrell, J. W., Kushnir, Y., Ottersen, G., & Visbeck, M. (2003). An overview of the North Atlantic Oscillation. In *Geophysical Monograph American Geophysical Union* (pp. 1–35). American Geophysical Union. <https://doi.org/10.1029/134GM01>

Le, T., & Bae, D.-H. (2020). Response of global evaporation to major climate modes in historical

and future Coupled Model Intercomparison Project Phase 5 simulations. *Hydrology and Earth System Sciences*, 24(3), 1131–1143. <https://doi.org/10.5194/hess-24-1131-2020>

Le, T., & Bae, D. (2022). Causal Impacts of El Niño–Southern Oscillation on Global Soil Moisture Over the Period 2015–2100. *Earth’s Future*, 10(3). <https://doi.org/10.1029/2021EF002522>

Le, T., Ha, K., & Bae, D. (2021). Increasing Causal Effects of El Niño–Southern Oscillation on the Future Carbon Cycle of Terrestrial Ecosystems. *Geophysical Research Letters*, 48(24). <https://doi.org/10.1029/2021GL095804>

Le, T., Kim, S., & Bae, D. (2022). Decreasing causal impacts of El Niño–Southern Oscillation on future fire activities. *Science of The Total Environment*, 826, 154031. <https://doi.org/10.1016/j.scitotenv.2022.154031>

McPhaden, M. J., Zebiak, S. E., & Glantz, M. H. (2006). ENSO as an integrating concept in earth science. *Science* (New York, N.Y.), 314(December), 1740–1745. <https://doi.org/10.1126/science.1132588>

Thirumalai, K., DInezio, P. N., Okumura, Y., & Deser, C. (2017). Extreme temperatures in Southeast Asia caused by El Niño and worsened by global warming. *Nature Communications*, 8, 1–8. <https://doi.org/10.1038/ncomms15531>

Reviewer #2 (Remarks to the Author):

REVIEWER REPORT

SUMMARY

In this study, the authors used a causal inference approach (Peter & Clark Momentary Conditional Independence, PCMCI) to evaluate the influence of atmospheric, hydrologic, and vegetation precursors on fires and their time lags.

GENERAL COMMENTS

GENERAL COMMENT #1:

My overall appreciation is that the study is very interesting and has the advantage of being performed at a global scale, but the manuscript has too many flaws and needs a major revision before can be accepted for publication. The most important problems are lack of clarity and detail, especially in what concerns data and methods; vague, obvious, redundant, and unnecessary sentences; conceptual and methodological issues. Another problem with this manuscript is that it is not clear what is its contribution to increasing knowledge about wildfire factors. The authors have to explain which of their results are new/innovative, i.e., what are the noteworthy results?

Next, I will address what seems to me to be the main problems with the study/manuscript and then list a set of specific questions/comments/suggestions that help explain my overall appreciation.

Response: We appreciate the reviewer's positive comment on this topic and regret our unclarity with the data, methods, and significance. We believe that our manuscript now has been improved significantly according to the reviewer's comments and suggestions. Our responses to the reviewer's general and specific comments are given below.

We also highlighted the noteworthy results in the Implications section:

In conclusion, this is the first globally consistent study partitioning top-down weather and bottom-up fuel precursors of wildfire activity. On a global scale, top-down weather precursors dominate in 73.3% of regions where causal relationships are detected. In the remaining 26.7% of regions, bottom-up fuel precursors are dominant, which coincides with North American and European boreal forests, and African and Australian savannahs. In these regions, fuel management practices may be the most efficient. (Line 267-271)

GENERAL COMMENT #2:

Confidence in the results and conclusions strongly depends on the quantity and quality of the data

as well as the adequacy of the methodology. The description of data and methodologies needs to be clarified and detailed. The reader does not always understand what the authors did, which prevents them from perceiving and validating results and conclusions. In this regard, one important issue is the air temperature (Temp) precursor. The authors do not explain which temperature (minimum, mean, maximum, other) and at what level (surface, 2 m, 10 m, 850 hPa, etc.). Based on the statement that “All the data sets are averaged or summed up to weekly...” maybe we can assume that Temp is some kind of average air temperature. However, it is also common knowledge that the vast majority of wildfires and burned areas occur under conditions more extreme than average. The authors somehow acknowledge this fact in the first paragraph of the "Main" section. Thus, in addition to the authors having to explain which temperature they used, if they used the mean temperature, they will have to explain why.

Response: We appreciate the reviewer’s constructive comment to clarify the Data section. We used daily mean 2m air temperature in the earlier submitted version. Thanks for the reviewer’s suggestion about the maximum temperature. We agree that wildfires are more related to the maximum temperature. So, in this revised version, we replaced the daily mean 2m air temperature with the daily maximum 2m air temperature. To make clear which variables/precursors were used in this study, we summarized this information in a new Supplementary Table (Table S1):

Table S1. All the indices involved in the causal analysis. The indices marked with ‘*’ were removed from this study according to the data selection result.

Group/Class	Index/Precursor	Calculated?	Product	Spatial Scale	Temporal Scale
Fire	Burned Area (BA)	-	Fire_CCI	250 m	Monthly
Top-down	Potential Evaporation (ET0)	-	ERA5	0.25 degree	Daily
	Aridity Anomaly Index (AAI)	√	(ERA5)	0.25 degree	Daily
	2m Maximum Air Temperature (Tmax)	-	ERA5	0.25 degree	Daily
	Surface Net Solar Radiation (Rad)*	-	ERA5	0.25 degree	Daily
	Vapor Pressure Deficit (VPD)	√	(ERA5)	0.25 degree	Daily
	Total Precipitation (Prec)*	-	ERA5	0.25 degree	Daily
	10m Wind Speed (Wind)	-	ERA5	0.25 degree	Daily
Bottom-up	Soil Water Deficit Index (SWDI)	√	(ERA5)	0.25 degree	Daily
	Fraction of Photosynthetically Active Radiation (FPAR)	-	MCD15A3H	500 m	4-day
	Normalized Difference Vegetation Index (NDVI)	-	MOD13C1	0.05 degree	16-day
	Enhanced Vegetation Index (EVI)	-	MOD13C1	0.05 degree	16-day
	Gross Primary Production (GPP)	-	MOD17A2HGF	500 m	8-day

GENERAL COMMENT #3:

The results of such a study strongly depend on the chosen precursors. A bad choice of precursors of a given type/group leads to equally bad results for that type of precursor. How can the reader be confident that the selected precursors are the best/most suitable? It is also not clear how the authors selected the precursors. Was there a pre-selection process? If yes, which one? How did they conclude

that four of each type would be the proper number?

Response: Thanks for the reviewer's comment. We performed an extensive study to evaluate the individual precursor contributions. We added a new section in the Supplementary Material to explain how we selected precursors:

Data Selection

Benefiting from the big data era (using remote sensing and reanalysis data), we can now perform a comprehensive wildfire causation study. We started by collecting all the data listed in **Table S1**. To make the comparison between the Top-down and Bottom-up groups more objective and transparent, we used a variable clustering method (“varclushi” package in Python) to execute the precursor selection.

Variable clustering is based on the Principal Component Analysis (PCA) algorithm. Firstly, all the input variables are started in one cluster, and a PCA is done for all the variables in this cluster. Secondly, the Eigenvalues (variance explained by each PC for all the variables) of the principal components are checked. The cluster is split if the second Eigenvalue is larger than a specified threshold (usually set as 0.7 or 1). This cluster-splitting process is iterated until all the second Eigenvalues in all clusters are lower than the specified threshold and the clusters cannot be further split. Thirdly, in each cluster, the representative variables having the highest correlation with their own cluster and the lowest correlation with other clusters are selected according to the formula below:

$$Ratio = \frac{1 - R_{own}^2}{1 - R_{nearest}^2}$$

where R_{own}^2 is the coefficient of determination (R^2) of the variable with its own cluster, and the $R_{nearest}^2$ is the R^2 of the variable with the nearest cluster. The one with the lowest *Ratio* is selected in each cluster.

Fig S1. Variable clustering result.

We used partial correlation in PCMCI to detect casual relationships, since it can eliminate the interactions between variables. As a consequence, all variables can be included in the PCMCI analysis. Considering that there are only five bottom-up precursors while seven top-down precursors, we decided to select the top five variables from the top-down group in order to prevent any artificial inflation of the importance of top-down variables. We applied variable clustering for all the 232 ecoregions and recorded the representative variables in each ecoregion. For each variable, we counted globally the pixels where it is representative, the result is shown in **Fig S1**. Therefore, according to the number of representative pixels, the variables that are finally selected were AAI, Wind, ET0, VPD, and Tmax in the Top-down group and SWDI, EVI, NDVI, GPP, and FPAR in the Bottom-up group. (*Supplementary Material Line 31-54*)

GENERAL COMMENT #4:

If I understand correctly, this study has serious conceptual (please see specific comments) and methodological flaws that can decisively affect the results and lead the reader to wrong conclusions. One of the most important conceptual/methodological problems is the names and composition of the precursors' groups. First, there are misclassified precursors. For example, the group of hydrological precursors does not seem to me to be the most suitable, as half of them are not hydrological, but atmospheric (precipitation and FWI). See examples referred to in specific comments. Secondly, and as a consequence of the first, the groups do not seem to me to be the most suitable. For example, in the case of hydrological precursors, it seems to me that it would be more appropriate to consider a group of soil precursors or, possibly, soil moisture precursors. On the one hand, the state and type of vegetation depend more on soil moisture than on all other hydrological aspects. I conclude that the authors agree with this statement because the group of hydrological precursors integrate measurements of soil moisture and not hydrology (e.g., surface and groundwater, with regard to their mechanical, physical and chemical properties, their geographic distribution, etc.). On the other hand, precursors should be independent within each group and between groups, which they clearly are not. The variables are all related to each other in such an obvious way that it is not even necessary to mention an example.

Response: We appreciate the reviewer's constructive comment, and we agree that such a grouping system could be misleading. In the revised version, we changed the grouping system. Now we categorize precursors into top-down and bottom-up groups, which is a widely used terminology in wildfire studies^{1,2}. We categorized Potential Evaporation (ET0), Aridity Anomaly Index (AAI), 2m Maximum Air Temperature (Tmax), Surface Net Solar Radiation (Rad), Vapor Pressure Deficit (VPD), Total Precipitation (Prec), and 10m Wind Speed (Wind) as top-down group, and categorized Soil Water Deficit Index (SWDI), Fraction of Photosynthetically Active Radiation (FPAR), Normalized Difference Vegetation Index (NDVI), Enhanced Vegetation Index (EVI), and Gross Primary Production (GPP) as bottom-up group. We believe this new grouping system is conceptually sound and in correspondence with existing fire literature.

We agree that it would be beneficial to ensure mutually independent. However, this would be a very stringent requirement. To eliminate the influence of interactions, we used partial correlation to detect casual relationships. Partial correlation quantifies the correlation between two variables when conditioned on other variables. After removing the effect of other variables, what remains is the partial correlation between the two target variables. Therefore, we can include all the wildfire precursors in this causation study, even if they are dependent; this is in fact the justification for and advantage of using PCMC³. To maintain a balance between the Top-down group and the Bottom-up group, we used variable clustering to find the representative precursors globally. By doing this, we cannot eliminate all interactions among precursors but try to keep them as independent as possible. By using the conceptual grouping system, variable clustering, and partial correlation, the approach allows attributing causation to the correct precursors even if they are interdependent.

Reference:

1. Walker XJ, Rogers BM, Veraverbeke S, Johnstone JF, Baltzer JL, Barrett K, et al. Fuel

availability not fire weather controls boreal wildfire severity and carbon emissions. Nature Climate Change 2020, 10(12): 1130-1136.

2. *Gill L, Taylor AH. Top-Down and Bottom-Up Controls on Fire Regimes Along an Elevational Gradient on the East Slope of the Sierra Nevada, California, USA. Fire Ecology 2009, 5(3): 57-75.*

3. *Runge J, Nowack P, Kretschmer M, Flaxman S, Sejdinovic D. Detecting and quantifying causal associations in large nonlinear time series datasets. Sci Adv 2019, 5(11): eaau4996.*

GENERAL COMMENT #5:

The wildfire research community recognizes and agrees that, for a vegetation wildfire to occur, the following requirements are necessary: (i) fuels/plant biomass, (ii) atmospheric conditions that lead the fuels/vegetation to an adequate state of dryness and, (iii) a source of ignition. The adequate state of dryness of the fuels (live or dead vegetation) is reached with atmospheric conditions of high temperature and low humidity that lead the vegetation to hydric and/or thermal stress for some time (depending on the type and state of the vegetation). Despite this common knowledge, the authors decided to reclassify atmospheric factors into atmospheric and hydrological. They consider: measures of energy (temperature, radiation, and evaporation) and call them atmospheric precursors; and, measures of water availability (precipitation, air, and soil humidity), which are essentially atmospheric and call them hydrological precursors. It is not possible to conclude that the hydrologic precursors are more or less important than the others if the hydrologic precursors are not hydrologic.

The authors also do not describe what might be the limitations and constraints that could affect the conclusions of the study.

Without a major revision of the manuscript to clarify and correct the problems identified in this general appreciation and in the specific comments, it is not worthwhile to analyze other aspects of the manuscript.

Response: Thanks for the reviewer's comment, we changed the grouping system as explained above. In the new system, we classify atmospheric conditions (including energy and humidity) as the top-down group, and fuel/plant conditions as the bottom-up group.

We also discussed the limitations of this study in the Implication part as follows:

There are three limitations of this study that could be addressed further. First, although benefiting from the big data era, the number of precursors included in this study is still limited. Major climate modes (e.g., El Niño–Southern Oscillation, and North Atlantic Oscillation) modulate regional top-down and bottom-up conditions and dominant wildfires^{69, 70}. Therefore, wildfire prediction could benefit from the achievable predictability of El Niño–Southern Oscillation⁷¹ and other major climate modes. Human behavior (e.g., ignition, cropland expansion, and fire suppression) is an important wildfire impact factor that may alter causal relationships. However, limited by the availability of high temporal resolution climate mode and human behavior indices, we currently cannot conduct such an analysis. Second, a better wildfire precursor grouping system

may be beneficial. While we divided the precursors into top-down and bottom-up groups which are conceptually clear⁷², sometimes, the top-down precursors dominate wildfires through their influence on fuel accumulation that is deduced from their long time lags, making the causal inference complex. Third, we assumed the causal relationship to be constant throughout the whole study period. However, extreme events or human behavior can alter fire regimes and potentially change causal relationships. (Line 253-266)

SPECIFIC COMMENTS

SPECIFIC COMMENT #1:

Line 4: In this manuscript, the authors used the concept of fire (a planned, wanted, and controlled combustion), which I believe is not correct; the authors should use the concept of wildfire (an unplanned and unwanted combustion of dead or alive vegetal biomass in a rural or natural area).

Response: We appreciate the accuracy of the word/term used by the reviewer and we have replaced fire with wildfire in the revision.

SPECIFIC COMMENT #2:

Line 4: the first sentence is completely vague; what role? What services? How?

Response: We deleted this sentence in the abstract.

SPECIFIC COMMENT #3:

Line 30: Is it really possible to "predict" wildfires "reliably and accurately?"

Response: Thanks for the reviewer's comment. We have rephrased this sentence to:

Better understanding and management of wildfires, therefore, is necessary to mitigate their negative impacts on human livelihood and the Earth's natural system. (Line 28-29)

SPECIFIC COMMENT #4:

Lines 31-32: "Fuel availability, usually in the form of vegetation conditions,...", fuel availability is not equal to vegetation conditions. Please correct the sentence.

Response: We appreciate the reviewer's careful review. We changed it to:

The availability of fuel is highly influenced by previous and current vegetation status and is a precondition for wildfires, yet extensive burning only occurs when fuels are dry and weather conditions are conducive to wildfire spread¹⁸. (Line 29-31)

SPECIFIC COMMENT #5:

Line 36: I agree that the FWI system is used widely. However, since it was developed for Canada, it must be calibrated before can be used in any other location. How did you calibrate the FWI indices

in order to be able to use them globally?

Response: Thanks for the reviewer's comment, we totally agree that the global use of the FWI system should be done in a careful manner. Its inputs (noon air temperature, air relative humidity, 24-hour accumulated precipitation, and wind speed) have become part of the Top-down group of precursors in the revision, while other direct measurements have been used to characterize bottom-up fuel status. To avoid this kind of conceptual problem, we have no longer included the FWI as a precursor in the revised analysis.

SPECIFIC COMMENT #6:

Line 46: I suppose that the black circle character is a typo/not necessary.

Response: Thanks for the comment, we checked the submitted .pdf file and the .pdf file downloaded from the submission system but did not find the black circle characters. We assume this is raised by a different kind or version of the .pdf reader. Apologies for this unclarity.

SPECIFIC COMMENT #7:

Line 46: you mentioned that you use air temperature (Temp). However, you do not explain what air temperature is; the minimum, mean, maximum or other air temperature? Air temperature at what level?

Response: Thanks for the reviewer's comment, we used maximum air temperature in the revised analysis. We listed all the precursors in a Table in the Supplementary Material as we responded in GENERAL COMMENT #2.

SPECIFIC COMMENT #8:

Line 48: can you explain why did you classify Precipitation, which can be defined as the fall of water, in any form (rain, snow, hail, hail, etc.), from the atmosphere to the earth's surface, as a hydrologic and not an atmospheric precursor?

Response: Thanks for the reviewer's comment. In the new grouping system, we classify Precipitation as a top-down precursor. However, the precursor pre-selection (variable clustering) showed its limited representativeness (Fig. S1), and as such precipitation was not included as one of the main precursors.

SPECIFIC COMMENT #9:

Lines 48-49: Please explain why you classify the fire WEATHER index (FWI) as a hydrologic and not an atmospheric precursor. Please note that the Canadian Forest Fire Weather Index (FWI) System consists of six components that account for the effects of fuel moisture and WEATHER conditions on fire behavior, computed with noon air temperature, air relative humidity and wind speed, and 24-hour accumulated precipitation.

Please quantify and explain how these misclassifications impact your findings.

Response: Thanks for the reviewer's comment. We abandoned FWI as it will raise a conceptual problem and we explained the reason above in SPECIFIC COMMENT #5:

"Its inputs (noon air temperature, air relative humidity, 24-hour accumulated precipitation, and wind speed) have become part of the Top-down group of precursors in the revision, while other direct measurements have been used to characterize bottom-up fuel status. To avoid this kind of conceptual problem, we have no longer included the FWI as a precursor in the revised analysis."

We also discussed the limitations of this study including the impact of the grouping system as we responded in GENERAL COMMENT #5:

Second, a better wildfire precursor grouping system may be beneficial. While we divided the precursors into top-down and bottom-up groups, which are conceptually clear⁷², sometimes, the top-down precursors dominate wildfires through their influence on fuel accumulation that is deduced from their long time lags, making the causal inference complex. (Line 260-264)

SPECIFIC COMMENT #10:

Line 57 (Fig. 1a): Please explain why the sum of the partial correlations of all 12 fire precursors per ecoregion ranges between 0 and 2.

Response: Thanks for the reviewer's question. The partial correlations are actually partial correlation coefficients (R), we summed the absolute value of partial correlation coefficients to find the hotspot regions in the submitted version. Now we realize that the partial coefficient of determination (R^2) is easier to understand, so we summed all the partial R^2 , in this case, the sum values should range between 0 and 1.

Fig 1. The sum of the partial coefficient of determination (partial R^2) of all 10 wildfire precursors per ecoregion, with higher values meaning higher explained variance of burned area, and vice versa.

SPECIFIC COMMENT #11:

Lines 57-61: "For all the 232 ecoregions, the dominant area fraction of the precursor (Fig. S1) shows

that VPD (47.6%) is the most important atmospheric precursor followed by Rad, Temp, and ET0 (23.7%, 18.1%, 10.6%, respectively)". What is the meaning of these results? Does this mean that VPD is the most important precursor in 47.6% of the 232 ecoregions, analyzed independently or jointly? If I understand correctly, the sum of the percentages of the four precursors of each type is 100%. Does this mean that you apply PCMCI independently for the precursors of each type/group (for all the ecoregions)? Please clarify.

Response: We appreciate the careful review. We only applied PCMCI once in each ecoregion and we may get a dominant precursor belonging to either the top-down or bottom-up group. Then we do the comparison in each group based on the dominant pixels. We changed the description as follows:

To investigate the dominant wildfire precursors globally, we applied PCMCI to build causal networks for each of the 232 ecoregions (29 climate zones multiplied by 8 vegetation types), using the top-down weather and bottom-up fuel grouping system. Then, in each ecoregion, we may detect a dominant precursor belonging to either the top-down or bottom-up group according to whether there is a causal relationship and the strength of the causal relationship (in this case partial correlation, Fig. 1). (Line 52-56)

We ranked the importance (dominant area fraction/pixel number) of the precursors in each group (Fig. S1). In the top-down group, ET0 is the most important precursor followed by VPD, AAI, Tmax, and Wind, accounting for 41.1%, 32.0%, 16.6%, 10.3%, and 0% of top-down dominant regions, respectively. In the bottom-up group, SWDI is the most important precursor followed by GPP, EVI, FPAR, and NDVI, accounting for 56.4%, 21.4%, 11.1%, 8.6%, and 2.5% of bottom-up dominant regions, respectively. (Line 64-69)

SPECIFIC COMMENT #12:

Lines 72-73: How did you compute these percentages? Did you use the number of ecoregions, the area, or what?

Response: We appreciate the reviewer's careful review. We used the dominant pixel numbers to compute these percentages and changed the description as:

Globally, the top-down group shows more dominance than the bottom-up group, accounting for 73.3% and 26.7 % of the globe where causal relationships are detected (Fig. 1c). (Line 59-61)

SPECIFIC COMMENT #13:

Line 83 and 91: this is only due to the precursors you selected and how you classify them.

Response: Thanks for the reviewer's comment. We believe that our revised top-down and bottom-up grouping system addresses this concern, and we revised the relevant statements as follows:

Wildfires are positively related to top-down precursors globally (91.6% positive, Fig. 1d). This positive relationship indicates that in flammability-limited regions, BA will increase when more

energy is supplied (higher Tmax and ET0) or the environment is drier (higher VPD and AAI). This is supported by evidence indicating that wildfires increase with the increased frequency and intensity of summer heatwaves²⁵, the strengthened solar radiation absorption^{26, 27}, and the increased atmospheric water demand and drought conditions^{28, 29} (see Fig. S2 for ET0, AAI, and VPD dominance). Major climate modes (e.g., El Niño–Southern Oscillation, and North Atlantic Oscillation) also modulate the occurrence of these weather anomalies^{30, 31, 32, 33, 34} and as such play an important role in governing global patterns of wildfire activities^{33, 35, 36}. The top-down dominance in the rainforest is supported by studies showing that here wildfires are limited to the time window when fuels are dry enough to burn^{37, 38}, see Fig. S2 for the AAI dominance. (Line 86-97)

SPECIFIC COMMENT #14:

Line 87: another black circle!

Response: Thanks for the comment, and apologies for this inconvenience which may be related to different pdf readers.

SPECIFIC COMMENT #15:

Lines 87-89: An obvious and unnecessary sentence. Is there any region where the atmospheric precursors do not play an important role?

Response: Thanks for the reviewer’s detailed suggestion to improve the statement of this manuscript. We revised the relevant statement as follows:

The top-down dominance in the rainforest is supported by studies showing that here wildfires are limited to the time window when fuels are dry enough to burn^{37, 38}, see Fig. S2 for the AAI dominance. (Line 95-97)

SPECIFIC COMMENT #16:

Lines 91-92: “Fires tend to be negatively related to hydrologic precursors (72.6%), and less frequently this relationship is positive (27.4%), see Fig. 1d”; the part “, and less frequently this relationship is positive (27.4%),” of the sentence is completely unnecessary; the reader can easily compute $100-72.6=27.4$.

Response: Thanks for the reviewer pointing this out. We revised the relevant statement as follows:

Wildfires are positively related to top-down precursors globally (91.6% positive, Fig. 1d). (Line 86-87)

Wildfires tend to be negatively related to bottom-up precursors (78.9% negative, Fig. 1d). (Line 98)

SPECIFIC COMMENT #17:

Lines 92-94: another obvious and unnecessary sentence; it is obvious that drought and precipitation deficits may provide optimal conditions for wildfires while the opposite, a “high water supply” can prevent fires.

Response: Thanks for the reviewer's comment. We deleted redundant and unnecessary sentences throughout the manuscript.

SPECIFIC COMMENT #18:

Lines 94-96: What “dominant positive relationship”?

Response: Thanks for the reviewer's comment. As we changed the grouping system, there is no more hydrologic group and this sentence has been removed from the revised manuscript. We checked the statements about partial correlation relationships (sign) and avoided such ambiguous statements.

SPECIFIC COMMENT #19:

Line 110: I am confused. If “All the data sets are averaged or summed up to weekly” (line 287), how can you have assessed the partial correlation for time lags shorter than 1 week?

Response: Thanks for the reviewer pointing this out and we regret such an inaccurate statement. We revised it as:

Longer time lags are shown in the boreal forests, African and Australian savannahs, the Indian Peninsula, the Indochina Peninsula, and southeastern South America (Fig. 2a). In the other regions, wildfires are a direct result of the precursing conditions (mostly no more than 1 week). (Line 111-113)

SPECIFIC COMMENT #20:

Lines 112-113: “..., which means in such regions fires are ignited shortly after (within a few days) the changes in the fire precursors.” another obvious and unnecessary sentence.

Response: Thanks for the reviewer's comment. We deleted such redundant sentences in the manuscript and made the statements more concise and meaningful as follows:

In the other regions, wildfires are a direct result of the precursing conditions (mostly no more than 1 week). (Line 112-113)

The bottom-up precursors can also come with negligible time lags (0 weeks, Fig. 2b), mainly in Australia and South America (Fig. 2c and Fig. S7), indicating that they can serve as not only fuel accumulation (longer time lag) but also fuel dryness (short time lag) indicators. (Line 127-130)

SPECIFIC COMMENT #21:

Lines 118-119: what evidence do you have for such a statement? That should not be the reason, at least everywhere.

Response: Thanks for the question. We cited a paper to support our explanation. A time lag of half a year is likely to follow from a peak in vegetation productivity and thus fuel accumulation in the wet season, which is then followed by fires in the subsequent dry season.

In this case, one possible reason is that the top-down precursors affect wildfires through their influence on fuel accumulation and availability. This is supported by a study conducted by Murray-Tortarolo et al., 2016⁴¹ indicating that in dry ecosystems, the dry season precipitation affects annual net primary productivity. (Line 118-121)

SPECIFIC COMMENT #22:

Lines 125-127: this does not seem correct/accurate; the tropics are a very large region; it is possible to be much more accurate. The explanation does not also seem correct. Please note that 30 weeks is less than a year.

Response: Thanks for pointing this out. We detailed the regions and revised the sentence as follows:

The bottom-up precursors show long time lags, especially in Europe, Asia, North America, and Africa (more than half a year, Fig. 2c). One possible reason could be that the current vegetation status or accumulated fuel is related to water supply and vegetation growth in the pre-wildfire seasons^{40, 42}. This is supported by studies showing that spring greening related to global warming can either be positively or negatively related to total biomass^{43, 44}, through the carryover effect of current vegetation states on subsequent growth caused by increased photosynthesis or earlier soil moisture depletion^{39, 40}. (Line 122-127)

We cited the “compensation effects of spring warming” related studies, especially reference 44, to support our analysis. In their study, they found that annual spring temperature affects subsequent summer and autumn NDVI. If we take the pre-growing season into account, this carryover effect would last longer from pre-growing to the whole growing season. This is also supported by Murray-Tortarolo et al. (2016).

Reference :

Buermann W, Forkel M, O'Sullivan M, Sitch S, Friedlingstein P, Haverd V, et al. Widespread seasonal compensation effects of spring warming on northern plant productivity. Nature 2018, 562(7725): 110-114.

Murray-Tortarolo G, Friedlingstein P, Sitch S, Seneviratne SI, Fletcher I, Mueller B, et al. The dry season intensity as a key driver of NPP trends. Geophysical Research Letters 2016, 43(6): 2632-2639.

SPECIFIC COMMENT #23:

Line 140 and 154: area of what?

Response: Thanks for the reviewer's careful review. We revised them as follows:

To generalize the previous analysis, we evaluated the dominance changes (according to the dominant pixel number) of top-down and bottom-up precursors that are primarily driven by energy,

dryness, vegetation, and location (Tmax, VPD, EVI, and latitude, Fig. 3). (Line 143-145)

We replaced the “dominant area fraction” with “dominant fraction” and explained how the dominant fraction is calculated.

The dominant fraction that is primarily driven by the group (top-down or bottom-up), is binned by temperature, AAI, NDVI, and latitude. The fractions were calculated by dividing the number of pixels where a specific precursor group is dominant by the number of pixels located in a specific bin and dominated by any of the two groups. (Line 159-162)

SPECIFIC COMMENT #24:

Line 154: another black circle.

Response: Thanks for the comment, and apologies for this inconvenience which may be related to different pdf readers.

SPECIFIC COMMENT #25:

Line 165: another black circle.

Response: Apologies for this inconvenience.

SPECIFIC COMMENT #26:

Line 196: in the Köppen-Geiger climate classification, the B type of climate is arid, not dry. Please correct.

Response: Thanks for pointing this out, we have corrected this in the revision.

SPECIFIC COMMENT #27:

Lines 254-270 (data): Apparently, you used monthly fire data can the reader assume that the time series of all other variables are also monthly? If not (line 287), how did you proceed to analyze data with different temporal sampling? Datasets do not have the same spatial resolution; how did you process the data with different spatial resolutions?

Response: Thanks for the question. We listed the original spatial and temporal scale of the data sets in the Supplementary Material, as explained in GENERAL COMMENT #2. We also modified the Data section for further clarification:

All the above-mentioned data sets span from 2003 to 2020. Since not the entire world surface has experienced wildfires in the past decades, we performed the analysis based on ecoregions instead of at a pixel level. In each of the 232 ecoregions (29 climate zones and 8 land cover classes), BA and precursors were resampled to a spatial resolution of 0.25° and averaged (precursors) or summed up (BA) to weekly values. Then, we calculated the ecoregional mean (precursors) or sum (BA) values by averaging or summing up all the pixels located in the ecoregions. (Line 301-306)

SPECIFIC COMMENT #28:

Lines 262-263: Why did you use temperature, radiation, and precipitation of the Global Land Data Assimilation System (GLDAS) but ET₀, VPD, FWI and AAI from ERA5? Why did you not use all weather data from the same source, for example, ERA5, which provides you with not only global but also consistent (homogeneous) data?

Response: We appreciate the reviewer's constructive data suggestions. In this revised manuscript, we followed your suggestion by consistently using data from ERA5 and MODIS.

SPECIFIC COMMENT #29:

Lines 271-285: the reader may not have access to all publications cited in the manuscript; this is why the methodology must be, eventually briefly but clearly described, which is not the case. In addition, the description of how AAI was computed is unclear; for example, "For each year, we calculated the difference between AI at a certain period and the normal AI for the corresponding period to calculate the anomaly, then the AAI was obtained."

Response: Thanks. We revised the Methods to make everything clear and listed the equations of how AAI, VPD, and SWDI are calculated in the Supplementary Material as follows:

We used a linear condition independence test, partial correlation, in PCMCI (both the condition selection and condition independence test stages) and set a significance level of 0.05 and a maximum time lag of 156 weeks. In each of the 232 ecoregions, we input wildfire behavior indicator (BA), Top-down precursors (Tmax, VPD, ET₀, Wind, and AAI), and Bottom-up precursors (FPAR, GPP, NDVI, EVI, and SWDI) and applied PCMCI to get causal networks showing causal links, directions, strengths, and time lags. In this study, we only focus on causal links and the time lags from wildfire precursors to BA. In each ecoregion, after getting the causal network, only the precursors that show causal links to BA are ranked by causal strength (partial correlation), and the precursor ranked first is called the dominant precursor, and its time lag is called the dominant time lag. (Line 332-343)

Precursor Calculation

Aridity Anomaly Index (AAI)

AAI was developed by the Indian Meteorological Department. It is a real-time drought index that describes vegetation water stress. In this study, AAI was calculated by dividing the difference between potential evaporation and actual evaporation by potential evaporation, the formula is shown below. The potential evaporation and actual evaporation are both from ERA5.

$$AAI = 100 \times \frac{ET_a - ET_0}{ET_0}$$

where ET_a is evaporation and ET₀ is potential evaporation.

Vapor Pressure Deficit (VPD)

VPD indicates the ability of the atmosphere to extract water from the environment by measuring the difference between the amount of water vapor in the air and how much water vapor it can hold when the air is saturated. In this study, VPD was calculated using minimum and maximum 2m air temperature and dewpoint temperature from ERA5. The formulas are shown below. All the temperature products used are from ERA5.

$$VPD = e_s - e_a$$

$$e_s = \frac{(e_{tmax} + e_{tmin})}{2}$$

$$e_a = 0.6108 \times \exp\left(\frac{17.27 \times T_{dew}}{T_{dew} + 237.3}\right)$$

$$e_{tmax} = 0.6108 \times \exp\left(\frac{17.27 \times T_{max}}{T_{max} + 237.3}\right)$$

$$e_{tmin} = 0.6108 \times \exp\left(\frac{17.27 \times T_{min}}{T_{min} + 237.3}\right)$$

where e_s is saturation vapor pressure and e_a is actual vapor pressure. T_{dew} is dewpoint temperature, T_{max} is maximum air temperature, and T_{min} is minimum air temperature.

Soil Water Deficit Index (SWDI)

SWDI is an agricultural drought monitoring index that uses soil moisture (SM), field capacity (FC), and wilting point (WP). The formula can be found below. Soil moisture used in this study is from ERA5 (volumetric soil water layer 1). Field capacity and wilting point are from Montzka et al., 2017¹. (Supplementary Material Line 7-30)

Anomaly Calculation

$$Anomaly = \frac{y - mean}{std}$$

where y is the value of a time series at a given time point; $mean$ is the mean of all values at the same time point in all cycles, in our case, it is the mean of all values for the same week in 18 years (2003–2022); std is the standard deviation derived from all values for the same week of the year across the 18 years. (Supplementary Material Line 55-59)

SPECIFIC COMMENT #30:

Lines 286-305 (Data Processing Procedures): you mentioned that you use monthly fire data (line 256 “...It is a global and monthly BA product”) and “All the data sets are averaged or summed up to weekly values...”. Please explain this apparent contradiction.

So, you produce weekly time series for each of the 28 climate zones (from Köppen-Geiger) and 8 land cover classes (from ESA CCI). The questions are:

What was the original temporal resolution of the downloaded data?

How did you proceed to obtain the time series for each of the 232 ecoregions 28 (or 29) climate zones and 8 vegetation types?

Response: Thanks for the comment. We revised the Data section and added several sentences regarding FireCCI as follows:

Note that even though FireCCI51 is a monthly dataset, it includes a layer recording the first fire detection day. Therefore, we computed the weekly summed BA from this daily information. For the precursors, we calculated the weekly mean from their original temporal scale. (Line 306-309)

We made a Table in the Supplementary Material to show the original temporal and spatial resolution of the downloaded data as listed in GENERAL COMMENT #2.

We also clarified how we obtained the time series in each of the 232 ecoregions as follows:

In each of the 232 ecoregions (29 climate zones and 8 land cover classes), BA and precursors were resampled to a spatial resolution of 0.25° and averaged (precursors) or summed up (BA) to weekly values. Then, we calculated the ecoregional mean (precursors) or sum (BA) values by averaging or summing up all the pixels located in the ecoregions. (Line 303-306)

SPECIFIC COMMENT #31:

Line 294: Please clarify what is a "... PC condition selection ..."; all acronyms must be clearly defined.

Response: Thanks for the comment. We revised the statement as follows.

It consists of two stages: a Peter & Clark (PC) condition selection stage and a Momentary Conditional Independence (MCI) test stage. (Line 317-318)

SPECIFIC COMMENT #32:

Lines 294-295: Please clarify what is "...an MCI conditional independence test ..."; all acronyms must be clearly defined.

Response: Thanks for the comment. Please see SPECIFIC COMMENT #31.

SPECIFIC COMMENT #33:

Line 295: "In the first stage, irrelevant conditions are removed"; what is an irrelevant condition?

Thanks for the reviewer's careful comment. We revised the relevant statement as follows:

In the first stage, for each variable, the irrelevant conditions that do not pass the iterative independence test are removed based on a Markov set discovery algorithm. In the next stage, a false-positive control was adapted to highly relevant interdependent conditions from the first stage. (Line 318-321)

SPECIFIC COMMENT #34:

Lines 296-298: "Two main assumptions of PCMCI are time-series stationarity and causal

sufficiency. The first assumption can be satisfied by calculating anomalies and de-trending the time series.” Once again, the authors do not adequately explain what leads to doubt. First, obviously, it is not by calculating anomalies that the series becomes stationary; just think that the trend does not disappear. Second, you don't explain what trend you remove and how. Did you just remove the trend over the entire study period? Did you check and correct the seasonality? This whole procedure must be properly explained.

Response: We appreciate the reviewer's comment. We first removed the entire period trend by fitting a linear regression and then calculated the anomaly to remove the seasonal cycle. By doing this, we ensured stationarity. We revised the relevant statements as follows:

We aimed to achieve (a) causal sufficiency by involving a wide range of wildfire precursors, and (b) time series stationarity by detrending and calculating anomalies. We used linear regression to remove the trend over the entire period and calculated anomalies to remove the seasonal cycle and alleviate temporal autocorrelation. The formula for the anomaly calculation can be found in the Supplementary Material. (Line 330-334)

REVIEWER COMMENTS

Reviewer #1 (Remarks to the Author):

General comments

The authors addressed my comments, and the manuscript has improved. However, several points remain unclear. I have further comments as below:

Major comments

- The authors confirmed that the analysis is based on ecoregional mean/summation. However, this approach does not fully exploit the available high-resolution data (i.e., 0.25x0.25 degree). It potentially leads to losing important information because the results have lower spatial resolution compared to the input data. It is probably fine to use the regional mean of the precursors as it is assumed that in the selected region, the climate conditions are similar for all grid points. However, using the summed BA might not be appropriate to represent the diverse fire conditions for all grid points/pixels. In this case, I think a more appropriate approach can be as follows: applying the causal analysis using the regional mean of precursors and BA data of all grid points. Other traditional approach is to have the computations for all grid points separately (i.e., at a selected grid point only climate and BA data of that grid points is used).
- Please clarify if the current approach using ecoregional mean/summation is the only available one?
- If there are other methods available (e.g., my suggestions above), which method is better (i.e., providing more detailed and accurate results)?
- The authors argued that they must use the regional mean/summation because not all grid points having BA data. This is not convincing because for these grid points with no fire, the obtained results are automatically no causal impacts (without computations).

Minor comments

- The range of the areas of the ecoregions needs to be provided.

Reviewer #2 (Remarks to the Author):

Dear authors,

I believe that you have satisfactorily answered the questions raised and proceeded with the adequate revision of the manuscript according to the suggestions provided in the revision report.

I recommend that the manuscript can be accepted for publication.

Reviewer #1 (Remarks to the Author):

General comments

The authors addressed my comments, and the manuscript has improved. However, several points remain unclear. I have further comments as below:

Response: We appreciate the positive comment of the reviewer on our revised manuscript. Our responses to the reviewer's comments are given below.

Major comments

- The authors confirmed that the analysis is based on ecoregional mean/summation. However, this approach does not fully exploit the available high-resolution data (i.e., 0.25x0.25 degree). It potentially leads to losing important information because the results have lower spatial resolution compared to the input data. It is probably fine to use the regional mean of the precursors as it is assumed that in the selected region, the climate conditions are similar for all grid points. However, using the summed BA might not be appropriate to represent the diverse fire conditions for all grid points/pixels. In this case, I think a more appropriate approach can be as follows: applying the causal analysis using the regional mean of precursors and BA data of all grid points. Other traditional approach is to have the computations for all grid points separately (i.e., at a selected grid point only climate and BA data of that grid point are used).
- Please clarify if the current approach using ecoregional mean/summation is the only available one?
- If there are other methods available (e.g., my suggestions above), which method is better (i.e., providing more detailed and accurate results)?
- The authors argued that they must use the regional mean/summation because not all grid points have BA data. This is not convincing because for these grid points with no fire, the obtained results are automatically no causal impacts (without computations).

Response: We appreciate the constructive comments and suggestions provided by the reviewer to improve our manuscript. We agree that there are other approaches for spatiotemporal data aggregation to perform this causal study, e.g., at a pixel scale by using original pixel BA and wildfire precursor values or using original pixel BA value and ecoregional mean wildfire precursor values.

In this study, we performed the causal analysis at an ecoregional scale because: (i) We expect both bottom-up fuel and top-down weather conditions to be relatively similar within an ecoregion as the areas within ecoregions are by definition comparable in climate, soil, geology, topography, and vegetation¹. (ii) We want to generate a wildfire information system like a “look-up table” based on the climate zone and vegetation type of a region. Such a “look-up table” provides basic insights needed to develop a wildfire warning/predicting system for regions (even if the region may be

smaller than a 0.25°x 0.25° pixel) where we don't have enough in-situ or field data. In this respect, our ecoregional scale study is more appropriate to generalize findings at a global scale than pixel scale study, yet still allows for sufficient spatial variation when analyzing global fire regimes.

There are several notable wildfire studies also at the ecoregional scale. For example, Jones et al., 2022² reviewed the grid and ecoregional scale trends and drivers of wildfires to relate their study to previously published grid-scale and (eco)regional-scale studies. In their study, they used three (eco)regional layers including 14 macroregions from the Global Fire Emissions Database (GFED), ten ecoregions, and countries. Turco et al., 2018³ studied wildfires in Mediterranean Europe in 44 ecoregions defined by both wildfire information and environmental zones. Walker et al., 2020⁴ studied the dominant drivers of boreal wildfire severity according to four ecoregions grouped from 417 field sites. Williams et al., 2019⁵ studied the impacts of anthropogenic climate change on wildfires in California based on four ecoregions defined by vegetation types, climate conditions, wildfire information, and population density. Alizadeh et al., 2023⁶ studied the trends of wildfire danger in mountain regions of the western US by averaging pixel scale data to 15 ecoregions. Littell et al., 2018⁷ assessed future wildfires in the western US in ecoregions defined by wildfire climate controls.

These studies further demonstrate that the ecoregional scale is an appropriate level for assessing bottom-up fuel and top-down weather influences on wildfire activity. To avoid confusion, we have rephrased our statement of the revision:

“We performed the causal analysis at the ecoregional level considering that top-down weather and bottom-up fuel conditions are relatively similar within the same ecoregion⁷⁶. We believe that this spatial aggregation is appropriate to generalize findings at a global scale, yet still allows for sufficient spatial variation when analyzing global fire regimes. In addition, we can generate a “look-up table” based on ecoregions that provide basic insights needed to develop a wildfire warning/predicting system for locations where we do not have enough in-situ or field data.” (Line 301-307)

Reference:

1. Hughes RM, Omernik JM. Ecological regions (ecoregions). In: Environmental Geology). Springer Netherlands (1999).
2. Jones MW, et al. Global and regional trends and drivers of fire under climate change. Reviews of Geophysics, (2022).
3. Turco M, et al. Exacerbated fires in Mediterranean Europe due to anthropogenic warming projected with non-stationary climate-fire models. Nat Commun 9, 3821 (2018).
4. Walker XJ, et al. Fuel availability not fire weather controls boreal wildfire severity and carbon emissions. Nature Climate Change 10, 1130-1136 (2020).

5. Williams AP, et al. Observed Impacts of Anthropogenic Climate Change on Wildfire in California. *Earth's Future* 7, 892-910 (2019).
6. Alizadeh MR, et al. Elevation-dependent intensification of fire danger in the western United States. *Nat Commun* 14, 1773 (2023).
7. Littell JS, McKenzie D, Wan HY, Cushman SA. Climate Change and Future Wildfire in the Western United States: An Ecological Approach to Nonstationarity. *Earth's Future* 6, 1097-1111 (2018).

Minor comments

- The range of the areas of the ecoregions needs to be provided.

Response: Thanks for the reviewer's comment. We added two figures in the Supplementary Material to show the range of the ecoregions according to climate zones and vegetation types:

Fig S16. Köppen-Geiger climate zone. The main climates include A (equatorial), B (arid), C (warm temperate), D (Snow), and E (polar). The precipitation types include W (desert), S (steppe), f (fully humid), s (summer dry), w (winter dry), and m (monsoonal). The temperature types include h (hot arid), k (cold arid), a (hot summer), b (warm summer), c (cool summer), d (extremely continental), F (polar frost), and T (polar tundra).

Fig S17. Vegetation type from the European Space Agency (ESA) Climate Change Initiative (CCI). The vegetation types include sparse vegetation, grassland, cropland, shrubland, tree cover broad-leaf evergreen (TBE), tree cover broad-leaf deciduous (TBD), tree cover needle-leaf evergreen (TNE), tree cover needle-leaf deciduous (TND).

Reviewer #2 (Remarks to the Author):

Dear authors,

I believe that you have satisfactorily answered the questions raised and proceeded with the adequate revision of the manuscript according to the suggestions provided in the revision report.

I recommend that the manuscript can be accepted for publication.

Response: We appreciate the reviewer's contributions to improve this manuscript.

REVIEWERS' COMMENTS

Reviewer #1 (Remarks to the Author):

The authors addressed my comments. I may recommend publication of this work.